# Analyzing and Improving Model Collapse in Rectified Flow Models

## Abstract

Generative models aim to produce synthetic data indistinguishable from real distributions, but iterative training on self-generated data can lead to *model collapse (MC)*, where performance degrades over time. In this work, we provide the first theoretical analysis of MC in Rectified Flow by framing it within the context of Denoising Autoencoders (DAEs). We show that when DAE models are trained on recursively generated synthetic data with small noise variance, they suffer from MC with progressive diminishing generation quality. To address this MC issue, we propose methods that strategically incorporate real data into the training process, even when direct noise-image pairs are unavailable. Our proposed techniques, including Reverse Collapse-Avoiding (RCA) Reflow and Online Collapse-Avoiding Reflow (OCAR), effectively prevent MC while maintaining the efficiency benefits of Rectified Flow. Extensive experiments on standard image datasets demonstrate that our methods not only mitigate MC but also improve sampling efficiency, leading to higher-quality image generation with fewer sampling steps.

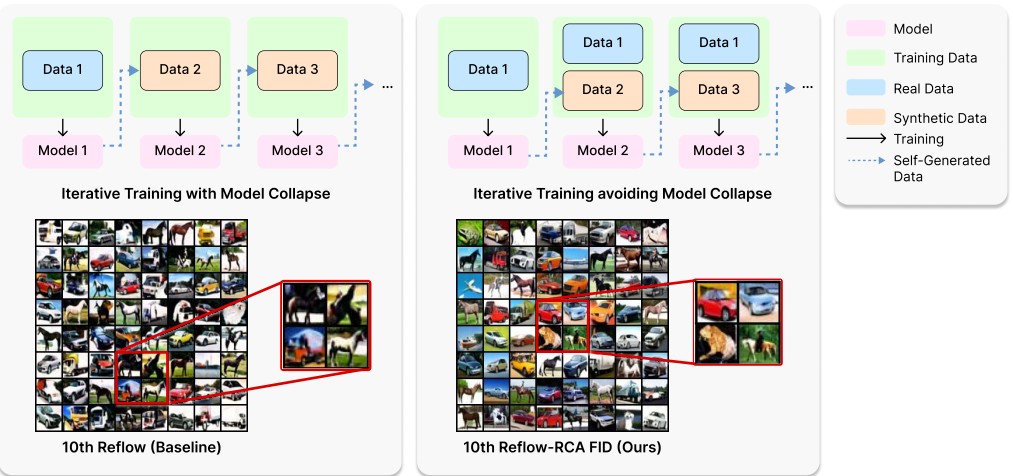

Figure 1: **Two Scenarios for Studying Model Collapse.** Top: MC occurs when successive iterations of generative models, trained on their own outputs, progressively degrade in performance, ultimately rendering the final model ineffective. Left: Illustrates MC by replacing data with each training iteration. Right: Depicts the scenario where original real data is added at each iteration, demonstrating that incorporating real data prevents the model from collapsing. Bottom: The correction streams trained in both modes after 10 iterations. The baseline lacks color, and the images are blurry and mixed. With our method, the images maintain their generated quality.

## 1 Introduction

Generative modeling aims to produce synthetic data that is indistinguishable from genuine data distributions. While deep generative models have achieved remarkable success across images, audio, and text (Rombach et al., 2022; Ramesh et al., 2022; Chen et al., 2020; Achiam et al., 2023; Touvron et al., 2023), the increasing reliance on synthetic data introduces significant challenges. A critical

| Methods | Variant | Performance | |
|---|---|---|---|
| | | **Efficient Sampling** | **Model Collapse Avoid** |
| Neural ODE/SDE | **Ours** | ✓ | ✓ |
| | DDPM (Ho et al., 2020) | ✗ | ✗ |
| | FM (Lipman et al., 2022) | ✓ (Weak) | ✗ |
| | OTCFM (Tong et al., 2023) | ✓ (Weak) | ✗ |
| | RF (Liu et al., 2022) | ✓ | ✗ |
| Distillation | CD/CT (Song et al., 2023) | ✓ (1-step) | Unknown |
| Collapse Avoid | MAD (Alemohammad et al., 2023) | ✗ | ✓ |
| | Stability (Bertrand et al., 2023) | ✗ | ✓ |
| | MCI (Gerstgrasser et al., 2024) | N/A | ✓ |
| | MCD (Dohmatob et al., 2024) | N/A | ✓ |

Table 1: Comparison of various methods regarding efficient sampling and model collapse avoidance. Symbols ✓ and ✗ indicate the presence or absence of a feature, respectively; "Weak" denotes limited capability, and "Unknown" or "N/A" indicates insufficient information or not applicable.

issue is *model collapse (MC)*, where generative models trained iteratively on their own outputs progressively degrade in performance (Shumailov et al., 2023). This degradation not only affects the quality of generated data but also poses risks when synthetic data is inadvertently included in training datasets, leading to self-consuming training loops (Alemohammad et al., 2023).

Simulation-free models and their variants—such as diffusion models (Song & Ermon, 2019; Song et al., 2020b; Ho et al., 2020), flow matching (Lipman et al., 2022; Pooladian et al., 2023; Tong et al., 2023), and Rectified Flow (Liu et al., 2022)—have drawn increasing attention. Among these models, Rectified Flow stands out due to its rapid development and extensive foundational and large-scale work (Esser et al., 2024). Unlike typical diffusion models, Rectified Flow's Reflow algorithm iteratively utilizes self-generated data as training data to straighten the flow and improve sampling efficiency, which closely aligns with the definition of MC. This direct use of self-generated data makes Rectified Flow an ideal candidate for studying and addressing MC. However, previous studies on Rectified Flow have primarily focused on scaling up the model or applying distillation techniques (Lee et al., 2024a; Liu et al., 2023; Esser et al., 2024), while neglecting a thorough analysis of MC itself. Consequently, the observed decline in Reflow's generation quality has been attributed to error accumulation without exploring the underlying mechanisms of collapse.

To address these challenges, we first delve into a theoretical analysis of MC in the case of Denoising Autoencoders (DAEs), then extend our investigation to Rectified Flow. We uncover the underlying mechanisms that lead to performance degradation when diffusion models and Rectified Flows are trained iteratively on their own outputs. Recognizing the limitations of previous approaches that primarily attribute decline to error accumulation, we aim to provide a deeper understanding of MC in this context. Building on this analysis, we propose novel methods to prevent MC in Rectified Flow. Our approaches involve the strategic incorporation of real data into the training process, even when direct noise-image pairs are not readily available. By leveraging reverse processes and carefully balancing synthetic and real data, we straighten the flow trajectories effectively while maintaining training stability. We validate our methods through extensive experiments on standard image datasets. The results demonstrate that our approaches not only mitigate MC but also enhance sampling efficiency, allowing for high-quality image generation with fewer sampling steps. This confirms the effectiveness of our strategies in both theoretical and practical aspects.

**Our main contributions are as follows: theoretical analysis:** To the best of our knowledge, we are the first to rigorously analyze model collapse in Rectified Flow and establish a theoretical framework using Denoising Autoencoders (DAEs). Specifically, we identify the causes of performance degradation due to iterative self-generated data training in Rectified Flow. We also introduce **novel methods** to prevent MC, including Reverse Collapse-Avoiding (RCA) Reflow, Online Collapse-Avoiding Reflow (OCAR), and OCAR-S, which preserve the efficiency of Rectified Flow while mitigating collapse. Moreover, we are the first to experimentally validate that the Reflow training method leads to a decrease in model performance, which suggests that most diffusion model distillation approaches that rely on synthetic data are also susceptible to MC. Finally, through **extensive experiments** on benchmark image datasets, we demonstrate that our methods effectively mitigate MC, improving both generation quality and sampling efficiency.

## 2 RELATED WORK

### 2.1 MODEL COLLAPSE IN GENERATIVE MODELS

The generation of synthetic data by advanced models has raised concerns about *model collapse*, where models degrade when trained on their own outputs. Although large language models and diffusion models are primarily trained on human-generated data, the inadvertent inclusion of synthetic data can lead to self-consuming training loops (Alemohammad et al., 2023), resulting in performance degradation (Shumailov et al., 2023). Empirical evidence of MC has been observed across various settings (Hataya et al., 2023; Martínez et al., 2023; Bohacek & Farid, 2023). Theoretical analyses attribute the collapse to factors like sampling bias and approximation errors (Shumailov et al., 2023; Dohmatob et al., 2024). While mixing real and synthetic data can maintain performance (Bertrand et al., 2023), existing studies often focus on maximum likelihood settings without directly explaining MC. Our work extends these analyses to simulation-free generative models like diffusion models and flow matching, specifically addressing MC in the Reflow method of Rectified Flow and proposing more efficient training of Rectified flow.

### 2.2 EFFICIENT SAMPLING IN GENERATIVE MODELS

Achieving efficient sampling without compromising quality is a key challenge in generative modeling. GANs (Goodfellow et al., 2014) and VAEs (Kingma & Welling, 2013) offer fast generation but face issues like instability and lower sample quality. Diffusion models (Song et al., 2020b) and continuous normalizing flows (Chen et al., 2018; Lipman et al., 2022; Albergo & Vanden-Eijnden, 2022), produce high-fidelity outputs but require multiple iterative steps, slowing down sampling. To accelerate sampling, methods such as modifying the diffusion process (Song et al., 2020a; Bao et al., 2021; Dockhorn et al., 2021), employing efficient ODE solvers (Lu et al.; Dockhorn et al., 2022; Zhang & Chen, 2022), and using distillation techniques (Salimans & Ho, 2022) have been proposed. Consistency Models (Song et al., 2023; Kim et al., 2023; Yang et al., 2024) aim for single-step sampling but struggle with complex distributions. Rectified Flow and its Reflow method (Liu et al., 2022; Lee et al., 2024b) promise efficient sampling by straightening flow trajectories, needing fewer steps. However, they are prone to MC due to training on self-generated data, and existing avoidance methods are ineffective as they do not provide the required noise-image pairs. Our work addresses this gap by proposing methods to prevent MC in Rectified Flow.

## 3 PRELIMINARIES

### 3.1 FLOW MATCHING

Flow Matching (FM) is a training paradigm for CNF (Chen et al., 2018) that enables simulation-free training, avoiding the need to integrate the vector field or evaluate the Jacobian, thereby significantly accelerating the training process (Lipman et al., 2022; Liu et al., 2022; Albergo & Vanden-Eijnden, 2022). This efficiency allows scaling to larger models and systems within the same computational budget. Let $\mathbb{R}^d$ denote the data space with data points $x \in \mathbb{R}^d$. The goal of FM is to learn a vector field $v_\theta(t, x) : [0, 1] \times \mathbb{R}^d \rightarrow \mathbb{R}^d$ such that the solution of the following ODE transports noise samples $x_0 \sim p_0$ to data samples $x_1 \sim p_1$:

$$\frac{d\phi_x(t)}{dt} = v_\theta(t, \phi_x(t)), \; \phi_x(0) = x. \tag{1}$$

Here, $\phi_x(t)$ denotes the trajectory of the ODE starting from $x_0$. FM aims to match the learned vector field $v_\theta(t, x)$ to a target vector field $u_t(x)$ by minimizing the loss:

$$\mathcal{L}_{\text{FM}}(\theta) = \mathbb{E}_{t \sim [0,1], x \sim p_t(x)} \|v_\theta(t, x_t) - u_t(t, x_t)\|_2^2, \tag{2}$$

where $p_t$ is the probability distribution at time $t$, and $u_t$ is the ground truth vector field generating the probability path $p_t$ under the marginal constraints $p_{t=0} = p_0$ and $p_{t=1} = p_1$. However, directly computing $u_t(x)$ and $p_t(x)$ is computationally intractable since they are governed by the continuity equation (Villani et al., 2009): $\partial_t p_t(x) = -\nabla \cdot (u_t(x)p_t(x))$.

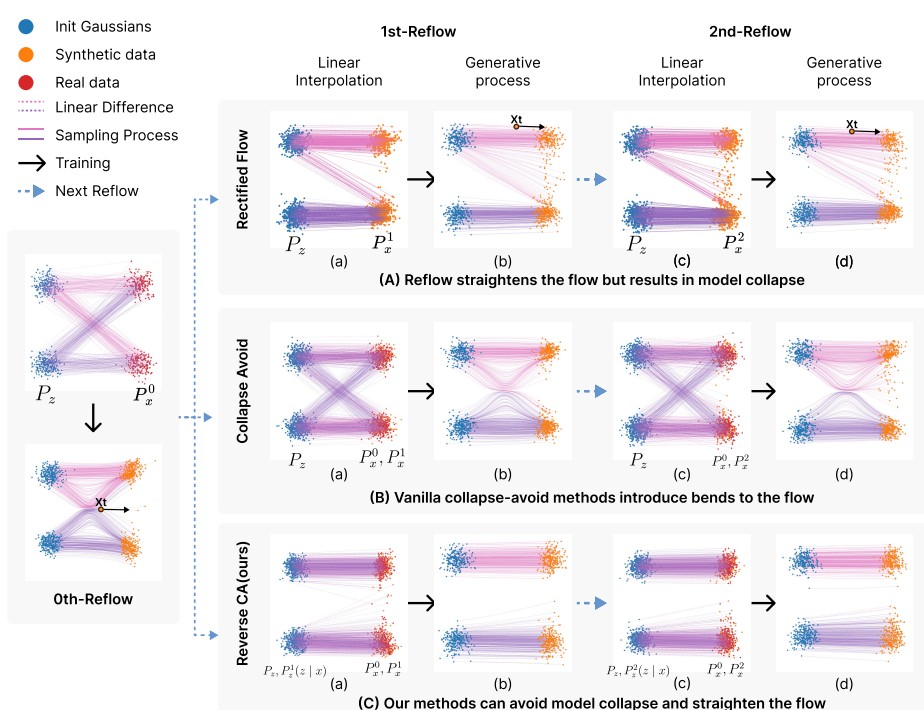

Figure 2: **2D multi-Gaussian experiment demonstration.** (A) Rectified Flow rewires trajectories to eliminate intersecting paths, transforming from (a) to (b). We then take noise samples from the distribution $p^z$ and their corresponding generated samples from the synthetic distribution $p_1^x$ to construct noise-target sample pairs (blue to orange) and linearly interpolate them at point (c). In Reflow, Rectified Flow is applied again from (c) to (c) to straighten the flows. This procedure is repeated recursively. (B) Since iterative training on self-generated data can cause MC, we can incorporate real data (shown in red) during training to prevent collapse. (C) However, adding real data introduces additional bends to the Rectified Flow because the pairs of real data and initial Gaussian samples are not pre-paired. Our method employs reverse sampling generated real-noise pairs (red to blue) to avoid MC while simultaneously straightening the flow.

To address this challenge, Lipman et al. (2022) proposes regressing $v_\theta(t, x)$ on a conditional vector field $u_t(x|z)$ and the conditional probability path $p_t(x|z)$, where $z \sim p(z)$ is an arbitrary conditioning variable independent of $x$ and $t$ (normally we set $p(z)$ as Gaussian Distribution).

$$\mathcal{L}_{\mathrm{CFM}}(\theta) = \mathbb{E}_{t \sim [0,1], z \sim p(z), x \sim p_t(x|z)} \left\| v_\theta(t, x_t) - u_t(t, x_t|z) \right\|_2^2. \tag{3}$$

Two objectives equation 2 and equation 3 share the same gradient with respect to $\theta$, while equation 3 can be efficiently estimated as long as the conditional pair $u(t, x_t|x), p_t(x_t|x)$ is tractable. By setting the $x_t = tz + (1-t)x$, $u(t, x_t|z) = \frac{z - x_t}{1-t}$ we get the loss of Rectified flow (Liu et al., 2022):

$$\mathcal{L}_{\mathrm{RF}}(\theta) = \mathbb{E}_{t \sim [0,1], z \sim p(z), x_1 \sim p_1} \left\| v_\theta(t, tz + (1-t)x) - (x - z) \right\|_2^2. \tag{4}$$

## 3.2 RECTIFIED FLOW AND REFLOW

Rectified Flow (RF) (Liu et al., 2022; Liu, 2022; Liu et al., 2023) extends FM by straightening probability flow trajectories, enabling efficient sampling with fewer function evaluations (NFEs). In standard FM, the independent coupling $p_{\boldsymbol{xz}}(\boldsymbol{x}, \boldsymbol{z}) = p_{\boldsymbol{x}}(\boldsymbol{x})p_{\boldsymbol{z}}(\boldsymbol{z})$ results in curved ODE trajectories, requiring a large number of function evaluations (NFEs) for high-quality samples. RF addresses this by iteratively retraining on self-generated data to rewire and straighten trajectories.

The *Reflow* algorithm (Liu et al., 2022) implements this idea by recursively refining the coupling between $\boldsymbol{x}$ and $\boldsymbol{z}$. Starting with the initial independent coupling $p_{\boldsymbol{x}_0\boldsymbol{z}}^{(0)}(\boldsymbol{x}, \boldsymbol{z}) = p_{\boldsymbol{x}_0}(\boldsymbol{x})p_{\boldsymbol{z}}(\boldsymbol{z})$, we can train the first Rectified flow $\theta_0$ by RF-loss equation 4 using stochastic interpolation data as input (see 2 0th-Reflow). Then, we can generate noise-image pairs because we can draw $(x_1, z)$ following

$dx_t = v_{\theta_0}(x_t, t)dt$ starting from $z \sim \mathcal{N}$ which means we can have $p_{\boldsymbol{x}_1\boldsymbol{z}}^{(1)}(\boldsymbol{x}, \boldsymbol{z}) = p_{\boldsymbol{x}_1}(\boldsymbol{x})p_{\boldsymbol{z}}(\boldsymbol{z})$ to start the reflow. Reflow generates an improved coupling $p_{\boldsymbol{x}\boldsymbol{z}}^{(k+1)}(\boldsymbol{x}_{k+1}, \boldsymbol{z})$ at each iteration $k$ by:

1. Generating synthetic pairs $(\boldsymbol{x}_k, \boldsymbol{z})$ sampled from the current coupling $p_{\boldsymbol{x}\boldsymbol{z}}^{(k)}(\boldsymbol{x}_k, \boldsymbol{z})$.
2. Training a new rectified flow $\theta_{k+1}$ by equation 4 using these synthetic pairs.

We denote the vector field resulting from the $k$-th iteration as the $k$-*Reflow*. This process aims to produce straighter trajectories, thus reducing the NFEs required during sampling. However, iterative training on self-generated data can cause *model collapse*, where performance degrades over iterations. Existing MC avoidance methods are ineffective for RF because incorporating real data does not provide the necessary noise-image pairs for Reflow training.

## 4 MODEL COLLAPSE ANALYSIS

### 4.1 CONNECTION BETWEEN DENOISING AUTOENCODERS AND DIFFUSION MODELS

Denoising Autoencoders (DAEs) are closely related to diffusion models through the concept of score matching (Song & Ermon, 2019; Song et al., 2020b). Under certain conditions, training a DAE implicitly performs score matching by estimating the gradient of the log-density of the data distribution (Vincent, 2011). Specifically, given data $\mathbf{x}$ and Gaussian noise $\epsilon \sim \mathcal{N}(0, \sigma^2 \mathbf{I})$, the DAE minimizes the reconstruction loss:

$$\mathcal{L}_{\text{DAE}} = \mathbb{E}_{\mathbf{x}, \epsilon} \left\| f_\theta(\mathbf{x} + \epsilon) - \mathbf{x} \right\|^2, \tag{5}$$

where $f_\theta$ is the DAE parameterized by $\theta$. The residual between the output and input approximates the scaled score function:

$$f_\theta(\mathbf{x} + \epsilon) - \mathbf{x} \approx \sigma^2 \nabla_{\mathbf{x}} \log p(\mathbf{x} + \epsilon). \tag{6}$$

We demonstrate that the training objectives of diffusion models and Flow Matching methods, such as Rectified Flow, can be unified, differing only in parameter settings and affine transformations (Esser et al., 2024). Specifically, diffusion models are special cases of Continuous Normalizing Flow (CNF) trajectories (Lipman et al., 2022; Liu et al., 2022). Consequently, analyzing MC in Denoising Autoencoders (DAEs) is essential for understanding collapse in diffusion models and Rectified Flow. Since DAEs learn to denoise and approximate the score function, examining their behavior under iterative training on self-generated data can reveal degradation mechanisms in more complex generative models. In this work, we focus on a simplified scenario where a DAE is recursively trained on its own generated data, enabling an analytical study of MC.

### 4.2 ANALYSIS OF DAE WITH RECURSIVELY LEARNING WITH GENERATIONAL DATA

To better understand the mechanisms behind MC, we investigate a simplified scenario where a linear DAE is trained recursively on the data it generates. Studying this setting provides valuable insights into how errors can accumulate over iterations, leading to performance degradation, which is challenging to analyze in more complex models.

Consider a two-layer neural network denoted by $f_\theta(\boldsymbol{x}) : \mathcal{X} \to \mathcal{X}$, which can be expressed in matrix form as $f_\theta(\boldsymbol{x}) = \boldsymbol{W}_2 \boldsymbol{W}_1 \boldsymbol{x}$, where $\boldsymbol{W}_2 \in \mathbb{R}^{d \times d'}$, $\boldsymbol{W}_1 \in \mathbb{R}^{d' \times d}$ represents the weights of one layer of the network, $\boldsymbol{\Phi} = \boldsymbol{W}_2 \boldsymbol{W}_1$. We aim to optimize the following training objectives:

$$\min_\theta \mathcal{L}(\theta) = \min_\theta \mathbb{E}_{\tilde{\boldsymbol{x}} \sim p(\boldsymbol{x}|\boldsymbol{z}), \boldsymbol{z} \sim \mathcal{N}} \left[ \left\| f_\theta(\tilde{\boldsymbol{x}}) - \boldsymbol{x} \right\|_2^2 \right], \tag{7}$$

where $\boldsymbol{z} \sim \mathcal{N}(0, \sigma^2)$ denotes Gaussian noise, $\boldsymbol{x} \sim p_1$ represents the original training data, and $\tilde{\boldsymbol{x}}$ is a perturbed version of $\boldsymbol{x}$, defined by $\tilde{\boldsymbol{x}} = \alpha \boldsymbol{x} + \beta \boldsymbol{z}$. The parameters $\alpha$ and $\beta$ are affine transformations that depend on the variable $t$. Here, we set $\alpha = \beta = 1$ for the simplicity of analysis. In practice, given a finite number of training samples, $\boldsymbol{X} = \begin{bmatrix} \boldsymbol{x}_1 & \cdots & \boldsymbol{x}_n \end{bmatrix}$, we learn the DAE by solving the following empirical training objectives

$$\theta^\star(\boldsymbol{X}) := \arg\min_\theta \sum_i \mathbb{E}_{\boldsymbol{z} \sim \mathcal{N}(0, \sigma^2 I)} \left[ \left\| f_\theta(\boldsymbol{x}_i + \boldsymbol{z}) - \boldsymbol{x}_i \right\|_2^2 \right], \tag{8}$$

where $\theta^\star(\boldsymbol{X})$ emphasizes the dependence of the solution on the training samples $\boldsymbol{X}$.

#### 4.2.1 Synthetic Data Generation Process

Now we formulate the Reflow of linear DAE. Suppose we have training data $\boldsymbol{X} = [\boldsymbol{x}_1 \quad \cdots \quad \boldsymbol{x}_n]$ with $\boldsymbol{x}_i = \boldsymbol{U}^\star \boldsymbol{U}^{\star\top} \boldsymbol{a}_i, \boldsymbol{a}_i \sim \mathcal{N}(0, I)$. Start with $\boldsymbol{X}_1 = \boldsymbol{X}$, in the $j$-th iteration with $j \geq 1$, the scheme for generating synthetic data is outlined as follows.

- Fit DAE: $(\boldsymbol{W}_2^j, \boldsymbol{W}_1^j) = \theta^\star(\boldsymbol{X}_j)$ by solving equation 8 with training data $\boldsymbol{X}_j$

- Generate synthetic data for the next iteration: $\boldsymbol{X}_{j+1} = \boldsymbol{W}_2^j \boldsymbol{W}_1^j (\boldsymbol{X}_j + \boldsymbol{E}_j)$, where each column of the noise matrix $\boldsymbol{Z}_j$ is iid sampled from $\mathcal{N}(0, \hat{\sigma}^2/n^2 I)$.

**Theorem 1.** *In the above synthetic data generation process 4.2.1, suppose that the variance of the added noise is not too large, i.e., $\hat{\sigma} \leq C\sigma$ for some universal constant $C$. Then, with probability at least $1 - 2je^{-n}$, the learned DAE suffers from MC as*

$$\|\boldsymbol{W}_2^j \boldsymbol{W}_1^j\|^2 \leq \frac{\|\boldsymbol{X}\|^2}{\sigma^2}\left(\frac{\|\boldsymbol{X}\|^2}{\|\boldsymbol{X}\|^2 + \sigma^2}\right)^{j-1}. \tag{9}$$

**Remark 1** (Connection to Diffusion Models)**.** *The primary gap between diffusion models and a sequence of end-to-end DAEs lies in the initial step of the diffusion process. This perspective aligns with discussions in Zhang et al. (2024), which examine the gap in the first step of diffusion models. For a detailed explanation, see Appendix A.2. The proof of Theorem 1 can be found in Appendix A.*

### 4.3 Does Model Collapse Occur in Rectified Flow?

Building on our analysis of MC in DAEs, we investigate whether a similar collapse occurs in Rectified Flow. Despite the differences between DAEs and Rectified Flow, we hypothesize that MC can still manifest in Rectified Flow when trained iteratively on self-generated data.

**Proposition 1.** *Let $v_{\theta_j}(t, \boldsymbol{x})_{j=1}^\infty$ be a sequence of vector fields trained via Reflow in Rectified Flow. As $j \to \infty$, due to the sampling process of Rectified Flow, the generated result $\boldsymbol{x}_{j,1}$ at time $t = 1$ (i.e., the output of the $j$-th Reflow iteration) converges to a constant vector, indicating model collapse.*

To test this hypothesis, we conducted experiments with Rectified Flow under iterative training. Our empirical results indicate that, without incorporating real data, the performance of Rectified Flow degrades over successive Reflow iterations, consistent with MC. For a detailed theoretical analysis and proof supporting this hypothesis, please refer to Appendix A.4.

### 4.4 Preventing Model Collapse by Incorporating Real Data

Incorporating real data into the training process is a strategy to prevent MC in generative models (Bertrand et al., 2023; Alemohammad et al., 2023; Gerstgrasser et al., 2024). Mixing real and synthetic data helps maintain performance and prevents degeneration caused by over-reliance on self-generated data. Inspired by these approaches, we extend the analysis of DEA by integrating real data. Recall the settings in 4.2.1, we modify the synthetic data generation scheme by adding real data. Specifically, we augment the current synthetic data with real data by setting $\hat{\boldsymbol{X}}_j = [\boldsymbol{X}_j \quad \boldsymbol{X}]$. To analyze the impact of adding real data, we present the following proposition (detailed settings and proof see Appendix A.3):

**Proposition 2.** *In the above synthetic data generation process 4.2.1 with adding real data, suppose that the variance of the added noise is not too large, i.e., $\hat{\sigma} \leq C\sigma$ for some universal constant $C$. Then, with probability at least $1 - 2je^{-n}$, the learned DAE does not suffer from model collapse as*

$$\|\boldsymbol{W}_2^j \boldsymbol{W}_1^j\|^2 \geq \frac{\|\boldsymbol{X}\|^2}{2\|\boldsymbol{X}\|^2 + \sigma^2}. \tag{10}$$

Compared to Theorem 1, Proposition A.1 shows that by incorporating real data into the synthetic data generation process, the learned DAE avoids model collapse, maintaining a fixed lower bound on the weight norm. In contrast, Theorem 1 indicates that without adding real data, the DAE's weight norm decreases exponentially with the number of iterations, leading to model collapse.

## 5 AVOIDING MODEL COLLAPSE IN RECTIFIED FLOW

Building upon our exploration of MC in simulation-free generative models, we address this challenge within the Rectified Flow framework. Although Rectified Flow and its Reflow algorithm (Liu et al., 2022) achieve efficient sampling by straightening probability flow trajectories, they are susceptible to MC due to iterative training on self-generated data (see Figure 2(A)). Our analysis, consistent with Bertrand et al. (2023); Gerstgrasser et al. (2024), shows that incorporating real data can mitigate collapse. However, integrating real data in Rectified Flow is challenging because it requires noise-image pairs that are not readily available, and directly pairing real images with random noise invalidates the Reflow training (see Figure 2(B)).

To overcome this limitation, we generate the necessary noise-image pairs using the reverse ODE process, commonly used in image editing tasks (Wallace et al., 2023; Zhang et al., 2023a). This allows us to obtain exact inverse image-noise pairs given the pre-trained model and real images. However, we face the issue of insufficient real image-noise pairs; for example, CIFAR-10 provides only 50,000 real images, while Reflow requires over 5 million data pairs per iteration (Liu et al., 2022). Our Gaussian experiments suggest that a synthetic-to-real data ratio of at least 7:3 is needed to avoid collapse effectively (see Figure 4). Using the reverse SDE process with significant randomness (Meng et al., 2021) leads to image-noise pairs dominated by randomness, undermining the purpose of straightening the flow (like the vanilla collapse-avoid methods Figure 2(B)).

Therefore, the question arises: *How can we generate sufficient real image-noise pairs while maintaining forward-backward consistency?* In the following sections, we detail the implementation of RCA, which effectively mitigates MC while preserving the efficiency benefits of Rectified Flow.

### 5.1 REVERSE COLLAPSE-AVOIDING REFLOW (RCA)

To address the challenge of generating sufficient real image-noise pairs while maintaining forward-backward consistency, we propose the **Reverse Collapse-Avoiding (RCA) Reflow** method. RCA Reflow leverages the reverse ODE process to generate noise-image pairs from real data. These real reverse pairs are then mixed with synthetic pairs obtained from the forward ODE process using a mix ratio $\lambda$, effectively mitigating MC without compromising the straightness of the flow trajectories. In RCA Reflow, we periodically regenerate all real reverse image-noise pairs after every $\alpha$ epochs to prevent overfitting to stale data and maintain training effectiveness. The combined dataset $(\boldsymbol{z}_j^{(i)}, \boldsymbol{x}_j^{(i)})$ consists of the mixed pairs, which are used to train the vector field $v_{\theta_j}$ for the $j$-th Reflow iteration. The training involves sampling noise vectors $\boldsymbol{z}^{(i)}$ from a standard normal distribution $\mathcal{N}(\boldsymbol{0}, \mathbf{I})$ and propagating them through the forward ODE to obtain synthetic images $\hat{\boldsymbol{x}}^{(i)}$. Simultaneously, we generate reverse image-noise pairs by propagating real images $\boldsymbol{x}^{(i)}$ backward through the reverse ODE to obtain corresponding noise vectors $\hat{\boldsymbol{z}}^{(i)}$:

$$\hat{\boldsymbol{x}}^{(i)} = \text{ODE}_{v_\theta}(0, 1, \boldsymbol{z}^{(i)}), \quad \hat{\boldsymbol{z}}^{(i)} = \text{ODE}_{v_\theta}(1, 0, \boldsymbol{x}^{(i)}). \tag{11}$$

Here, we define the first-order Explicit Euler ODE sampler as $\text{ODE}_{v_\theta}(t_0, t_1, \boldsymbol{x}) : [0, 1] \times [0, 1] \times \mathbb{R}^d \to \mathbb{R}^d$, where $v_\theta$ is the trained Rectified Flow. To create a balanced and diverse training dataset, we mix the synthetic and real reverse pairs based on the mix ratio $\lambda$. Specifically, given $n$ synthetic pairs $\{(\boldsymbol{z}^{(i)}, \hat{\boldsymbol{x}}^{(i)})\}_{i=1}^n$ and $n$ real reverse pairs $\{(\hat{\boldsymbol{z}}^{(i)}, \boldsymbol{x}^{(i)})\}_{i=1}^n$, we randomly select $\lambda n$ synthetic pairs and $(1 - \lambda)n$ real reverse pairs to form the combined dataset $\mathcal{D}_j$:

$$\mathcal{D}_j = \left\{ (\boldsymbol{z}_j^{(i)}, \boldsymbol{x}_j^{(i)}) \right\}_{i=1}^n = \left\{ (\boldsymbol{z}^{(i)}, \hat{\boldsymbol{x}}^{(i)}) \right\}_{i=1}^{\lambda n} \cup \left\{ (\hat{\boldsymbol{z}}^{(i)}, \boldsymbol{x}^{(i)}) \right\}_{i=1}^{(1-\lambda)n} \tag{12}$$

This method ensures that the combined dataset comprises both synthetic and real reverse image-noise pairs, maintaining data diversity and preventing MC by leveraging the strengths of both data sources. We provide the detailed training procedure in Algorithm 1, which clarifies the steps involved in RCA Reflow.

---

**Algorithm 1** Reverse Collapse-Avoiding Reflow Training

---

**Require:** Reflow iterations $\mathcal{J}$; real dataset $\{\boldsymbol{x}^{(i)}\}$; pre-trained vector field $v_{\theta_0}$; mix ratio $\lambda$; ODE
   solver $\text{ODE}(t_0, t_1, \boldsymbol{x})$; regeneration parameter $\alpha$.
**Ensure:** Trained vector fields $\{v_{\theta_j}\}_{j=1}^{\mathcal{J}}$
  1: **for** $j = 1$ to $\mathcal{J}$ **do**
  2:    Sample $\{\boldsymbol{z}^{(i)}\}$ from $\mathcal{N}(\boldsymbol{0}, \mathbf{I})$
  3:    Compute $\hat{\boldsymbol{x}}^{(i)} = \text{ODE}(0, 1, \boldsymbol{z}^{(i)})$                    ▷ *Generate synthetic noise-image pairs*
  4:    Compute $\hat{\boldsymbol{z}}^{(i)} = \text{ODE}(1, 0, \boldsymbol{x}^{(i)})$          ▷ *Generate reverse image-noise pairs from real data*
  5:    Randomly select $\lambda n$ synthetic pairs and $(1 - \lambda)n$ real reverse pairs
  6:    $\mathcal{D}_j = \{(\boldsymbol{z}_j^{(i)}, \boldsymbol{x}_j^{(i)})\}_{i=1}^n = \{(\boldsymbol{z}^{(i)}, \hat{\boldsymbol{x}}^{(i)})\}_{i=1}^{\lambda n} \cup \{(\hat{\boldsymbol{z}}^{(i)}, \boldsymbol{x}^{(i)})\}_{i=1}^{(1-\lambda)n}$ ▷ *Mix Pairs with Ratio $\lambda$*
  7:    **repeat**                                                                 ▷ *Reflow training*
  8:       **for** each $(\boldsymbol{z}_j^{(i)}, \boldsymbol{x}_j^{(i)}) \in \mathcal{D}_j$ **do**
  9:          Sample $t \sim \mathcal{U}(0, 1)$
 10:          Compute $\boldsymbol{x}_t^{(i)} = t\,\boldsymbol{x}_j^{(i)} + (1 - t)\,\boldsymbol{z}_j^{(i)}$
 11:          Compute loss:

$$\mathcal{L}_{\text{RF}} = \frac{1}{B} \sum_{i=1}^{B} \left\| v_{\theta_j}(t, \boldsymbol{x}_t^{(i)}) - (\boldsymbol{x}_j^{(i)} - \boldsymbol{z}_j^{(i)}) \right\|^2$$

 12:          Update $\theta_j$ using gradient descent
 13:       **if** $j \mod \alpha = 0$ **then**                              ▷ *Re-generate pairs every $\alpha$ epochs*
 14:          Repeat Steps 2 and 4
 15:    **until** converged
 16: **Output**: $\{v_{\theta_j}\}_{j=1}^{\mathcal{J}}$

---

## 5.2 ONLINE REVERSE COLLAPSE-AVOIDING REFLOW (OCAR)

Although the Reverse Collapse-Avoiding (RCA) Reflow method effectively prevents model collapse and straightens the flow to reduce the sampling steps in Rectified Flow (Figure 2 (C)), it requires substantial storage resources. For instance, Lee et al. (2024a) report using over 40 GB of memory on the ImageNet $64 \times 64$ task just to store $\hat{\boldsymbol{x}}^{(i)}$ during one Reflow iteration. This high memory consumption limits the applicability of RCA in high-dimensional image generation experiments.

To address this limitation, we consider the scenario where the regeneration parameter $\alpha$ approaches zero. In this case, we obtain an online method that generates synthetic noise-image pairs and real reverse image-noise pairs in every mini-batch, mixing them on the fly. This approach resembles the distillation method proposed by Kim et al. (2023), who use real data to improve the performance of consistency models (Song et al., 2023). However, there are key differences: **First**, we do not use a fixed pre-trained model as a teacher; instead, we straighten the flow through repeated iterations. **Second**, we do not rely on the assumption that the neural network can recover any sampled point from any distribution on the generative path given any input. Instead, we maintain a straight path that is easy to understand and has clear meaning. Research shows that a straight flow can be regarded as a progressive approximation of the optimal transport map (Liu, 2022). The detailed algorithm for the **Online Collapse-Avoiding Reflow (OCAR)** method can be found in Appendix B.1.

## 5.3 DOES ADDING RANDOMNESS HELP? REVERSE SDE SAMPLING

In the previous methods, we utilized the reverse ODE process to generate noise-image pairs for training. However, relying solely on the deterministic ODE means that the only source of randomness in the training process comes from the initial latent variables $\boldsymbol{z} \sim \mathcal{N}(\boldsymbol{0}, \mathbf{I})$. This limited randomness may impact the diversity of generated samples and the robustness of the model (Zhang et al., 2023b).

To enhance diversity and potentially improve generation quality, we introduce controlled randomness into the reverse process by employing a reverse SDE. In practice, we set the noise scale $\sigma$ to be small (e.g., $\sigma = 0.001$) and perform sampling using methods like the Euler-Maruyama scheme with an appropriate number of steps (e.g., $leq 100$ steps). This controlled injection of noise increases vari-

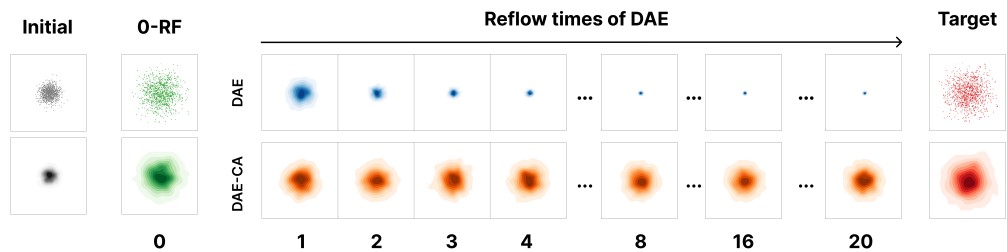

Figure 3: Reflow Process of the DAE on a 4-D Gaussian Distribution. The figure visualizes a slice of the distribution along dimensions 0 and 1. Both kernel density estimation plots and sample points are shown for the initial and target distributions.

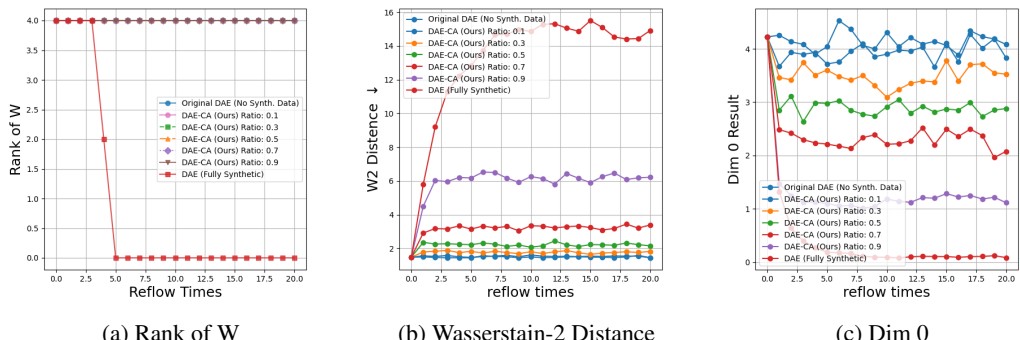

(a) Rank of W          (b) Wasserstain-2 Distance          (c) Dim 0

Figure 4: Results from the reflow experiment with DAE on 4D Gaussian.

ability without significantly disrupting the straightening effect of the flow. We denote this method as **OCAR-S**. More detail can be find in Appendix B.2

## 6 EXPERIMENTS

In this section, we first validate our analysis of model collapse in DAEs and its extension to diffusion models and Rectified Flow. We then demonstrate that our proposed methods—Reverse Collapse-Avoiding Reflow (RCA), Online Collapse-Avoiding Reflow (OCAR), and OCAR with added Stochasticity (OCAR-S)—are capable of producing high-quality image samples on several commonly used image datasets. Additionally, we show that our methods provide a more efficient straightening of the sampling path, allowing for fewer sampling steps on CIFAR-10 (Krizhevsky et al., 2009). Moreover, we demonstrate high-quality image generation on high-resolution datasets such as CelebA-HQ 256 (Karras, 2017), combined with latent space methods (Rombach et al., 2022) commonly used in Rectified Flow (Dao et al., 2023; Esser et al., 2024). We compare results using the Wasserstein-2 distance (W2, (Villani et al., 2009), lower is better), Fréchet Inception Distance (FID, (Heusel et al., 2017), lower is better), and the number of function evaluations (NFE, lower is better). Due to limited space, we place the further settings of each experiment in the appendix.

### 6.1 GAUSSIAN TASK

The intermediate columns of Figure 3 illustrate the progression of the DAE Reflow process at different stages. They demonstrate that the original DAE Reflow leads to model collapse, whereas our proposed collapse-avoiding DAE Reflow maintains the integrity of the generated data.

Figure 4 presents the key results from our DAE Reflow experiment on the 4D Gaussian task. The findings demonstrate that adding real data effectively prevents model collapse and maintains the integrity of the generated data. Specifically, incorporating real data helps maintain the rank of the weight matrix $W$ across Reflow iterations, our collapse-avoiding method consistently achieves a lower Wasserstein-2 distance compared to the original DAE Reflow, and the stability of the first principal component in PCA shows that our method effectively preserves the data structure over iterations More details can be fund in Appenxix C.1

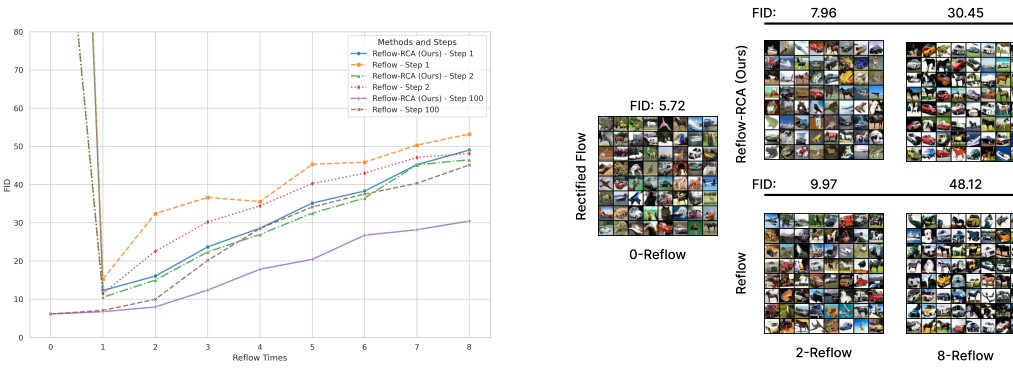

(a) Comparison of methods          (b) Samples demonstrating model collapse

Figure 5: **Comparison of Reflow and Reflow-RCA** We set $\lambda = 0.5, \alpha = 8$ and use a half-scale U-Net for the experiment. See Appendix 7 for full samples for reflow processing

| | CIFAR10 ($32 \times 32$) | | | | CelebA-HQ ($256 \times 256$) | | | |
|---|---|---|---|---|---|---|---|---|
| | 10 NFE | 20 NFE | 50 NFE | Best NFE | 10 NFE | 20 NFE | 50 NFE | Best NFE |
| 0-RF (ICFM) | **14.16** | 9.88 | 6.30 | 4.02/152 (**2.58/127**) | – | – | – | – |
| FM | 16.00 | 10.70 | 7.76 | 6.12/158 (6.35/142) | 16.51 | 8.40 | 5.87 | 5.45/89(5.26/89) |
| OTCFM | 14.47 | **9.38** | **5.78** | **3.96/134** (3.58/134) | – | – | – | – |
| 1-RF | 10.83 | 9.75 | 7.49 | 5.95/108 (3.36/110) | 12.04 | 7.34 | 5.76 | 5.73/71 |
| 1-RF-RCA (**Ours**) | **8.68** | **7.47** | **6.98** | **5.61/112** | **11.39** | **7.27** | **5.61** | **5.57/69** |
| 2-RF | 14.97 | 12.01 | 10.13 | 9.68/107(3.96/104) | 13.27 | 8.71 | 7.05 | 6.28/67 |
| 2-RF-RCA (**Ours**) | **11.47** | **9.12** | **8.58** | **7.64/102** | **12.89** | **8.50** | **6.91** | **6.10/67** |
| OCRA (**Ours**) | **7.02** | 6.30 | 5.96 | 4.27/96 | 10.89 | 7.12 | 5.60 | 5.52/69 |
| OCRA-S (**Ours**) | 7.45 | **6.01** | **5.19** | **4.15/94** | **10.86** | **6.99** | **5.53** | **5.49/70** |

Table 2: **Comparison of model collapse avoidance methods on FID score ($\downarrow$) for unconditional generation.** We set $\lambda = 0.5$, $\alpha = 2$, and use full-scale U-Net for CIFAR-10 and DiT-L/2 for CelebA-HQ. "Best NFE" is shown as "FID/NFE" using the DOPRI5 solver. Parentheses indicate results from the original papers. Variations may exist due to different neural network settings or random seeds; however, the comparison remains fair.

## 6.2 Straight Flow and Fewer-Step Image Generation

**Reverse Collapse-Avoiding Reflow** Our experiments on CIFAR-10 show that Reflow achieves more efficient flows, enabling the use of fewer sampling steps. As illustrated in Figure 5(a), we observe the following key findings: **First**, 0-Reflow (vanilla Rectified Flow or FM) fails to enable 1- or 2-step sampling, whereas 1-RF and larger variants of RF can; **Second**, our RCA Reflow method effectively prevents model collapse, resulting in more efficient training, as shown in Figure 5(b). Specifically, Table 2 demonstrates that Rectified Flow trained with RCA Reflow generates high-quality images using only a few sampling steps, underscoring the improvement in flow straightness. Detailed experimental settings and additional ablation studies can be found in Appendix C.3.

**Online Collapse-Avoiding Reflow.** RCA Reflow can be considered a pseudo-online method. For OCAR, we employ a full-size U-Net using the same settings as in Lipman et al. (2022); Dao et al. (2023). As shown in Table 2, both OCRA and OCRA-S outperform vanilla Reflow, achieving better FID scores than our RCA method, without requiring additional storage.

## 7 Conclusion

We addressed model collapse in simulation-free generative models, focusing on Rectified Flow. Through theoretical analysis, we identified how training on self-generated data leads to performance degradation. To mitigate this, we introduced RCA Reflow and OCAR, which incorporate real data to prevent collapse while maintaining efficiency. Experiments validate their effectiveness in improving generation quality and sampling efficiency. Future work includes exploring model collapse in other distillation methods, such as Consistency Distilling (CD), and further enhancing the robustness of generative models under more challenging conditions.

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

# Appendix

CONTENTS

# A PROOFS AND FORMULATIONS

## A.1 PROOF OF THEOREM 1

*Proof of Theorem 1.* We can first expand the training loss in equation 8 as follows:

$$\mathcal{L}(\theta) = \mathbb{E}_{\boldsymbol{z} \sim \mathcal{N}(0, \sigma^2 I)} \left[ \|\boldsymbol{W}_2 \boldsymbol{W}_1 \boldsymbol{x}_i - \boldsymbol{x}_i\|^2 - 2\langle \boldsymbol{W}_2 \boldsymbol{W}_1 \boldsymbol{x}_i - \boldsymbol{x}_i, \boldsymbol{W}_2 \boldsymbol{W}_1 \boldsymbol{z} \rangle + \|\boldsymbol{W}_2 \boldsymbol{W}_1 \boldsymbol{z}\|_2^2 \right],$$

$$= \sum_{i=1}^{n} \|\boldsymbol{W}_2 \boldsymbol{W}_1 \boldsymbol{x}_i - \boldsymbol{x}_i\|^2 + \sigma^2 \|\boldsymbol{W}_2 \boldsymbol{W}_1\|_F^2. \tag{13}$$

We denote by $\boldsymbol{\Phi} = \boldsymbol{W}_2 \boldsymbol{W}_1$ to simplify the following analysis. The induced $\ell_2$ regularization in equation 13 suggests that DAE performs denoising by learning a low-dimensional model. The optimal solution for equation 13, written in terms of $\boldsymbol{\Phi}$, is simply given by $(\boldsymbol{X}\boldsymbol{X}^\top)(\boldsymbol{X}\boldsymbol{X}^\top + \sigma^2 I)^{-1}$. When $\sigma \to 0$, the solution converges to PCA. Plugging this into the process of recursively learning DAE from generational data, we have $\boldsymbol{\Phi}_j = \boldsymbol{W}_2^j \boldsymbol{W}_1^j = (\boldsymbol{X}_j \boldsymbol{X}_j^\top)(\boldsymbol{X}_j \boldsymbol{X}_j^\top + \sigma^2 I)^{-1}$.

Let $\lambda(\cdot)$ denote the largest eigenvalue of a matrix. Since $\boldsymbol{X}_{j+1} = \boldsymbol{\Phi}_j(\boldsymbol{X}_j + \boldsymbol{E}_j)$ with each column of $\boldsymbol{E}_j$ being iid sampled from $\mathcal{N}(0, \hat{\sigma}^2/n^2 I)$, it follows from (Vershynin, 2018, Theorem 4.6.1) that there exists a constant $C$ such that, with probability at least $1 - 2e^{-n}$, $\lambda(\boldsymbol{X}_{j+1}\boldsymbol{X}_{j+1}^\top) \leq \lambda^2(\boldsymbol{\Phi}_j)(\lambda(\boldsymbol{X}_j\boldsymbol{X}_j^\top) + C\hat{\sigma}^2)$. This together with $\lambda(\boldsymbol{\Phi}_j) = \frac{\lambda(\boldsymbol{X}_j\boldsymbol{X}_j^\top)}{\lambda(\boldsymbol{X}_j\boldsymbol{X}_j^\top) + \sigma^2}$ implies that when $\hat{\sigma}^2 \leq \sigma^2/C$,

$$\lambda(\boldsymbol{X}_{j+1}\boldsymbol{X}_{j+1}^\top) \leq \lambda(\boldsymbol{X}_j\boldsymbol{X}_j^\top)\lambda(\boldsymbol{\Phi}_j)\frac{\lambda(\boldsymbol{X}_j\boldsymbol{X}_j^\top) + C\hat{\sigma}^2}{\lambda(\boldsymbol{X}_j\boldsymbol{X}_j^\top) + \sigma^2} \leq \lambda(\boldsymbol{X}_j\boldsymbol{X}_j^\top)\lambda(\boldsymbol{\Phi}_j) \tag{14}$$

holds with probability at least $1 - 2e^{-n}$. Denote by $\tau = \lambda(\boldsymbol{X}\boldsymbol{X}^\top)$. In the following, we prove that with probability at least $1 - 2qe^{-n}$,

$$\left[ \lambda(\boldsymbol{X}_q\boldsymbol{X}_q^\top) \right] \leq \lambda(\boldsymbol{X}\boldsymbol{X}^\top)(\frac{\tau}{\tau + \sigma^2})^{q-1}. \tag{15}$$

We prove this by induction. It holds when $q = 0$. Now assume equation 15 is true at $q = j$. We prove it also holds at $q = j + 1$. Since equation 15 holds at $j$, we have $\lambda(\boldsymbol{X}_j\boldsymbol{X}_j^\top) \leq \lambda(\boldsymbol{X}\boldsymbol{X}^\top)$, and hence $\lambda(\boldsymbol{\Phi}_j) = \frac{\lambda(\boldsymbol{X}_j\boldsymbol{X}_j^\top)}{\lambda(\boldsymbol{X}_j\boldsymbol{X}_j^\top) + \sigma^2} \leq \frac{\tau}{\tau + \sigma^2}$. Plugging this into equation 14 gives

$$\lambda(\boldsymbol{X}_{j+1}\boldsymbol{X}_{j+1}^\top) \leq \lambda(\boldsymbol{X}_j\boldsymbol{X}_j^\top)\lambda(\boldsymbol{\Phi}_j) \leq \lambda(\boldsymbol{X}\boldsymbol{X}^\top)(\frac{\tau}{\tau + \sigma^2})^j.$$

This proves equation 15. Finally, we can obtain the bound for $\lambda(\boldsymbol{\Phi}_j)$ as

$$\lambda(\boldsymbol{\Phi}_j) = \frac{\lambda(\boldsymbol{X}_j\boldsymbol{X}_j^\top)}{\lambda(\boldsymbol{X}_j\boldsymbol{X}_j^\top) + \sigma^2} \leq \frac{\lambda(\boldsymbol{X}\boldsymbol{X}^\top)(\frac{\tau}{\tau+\sigma^2})^{j-1}}{\lambda(\boldsymbol{X}\boldsymbol{X}^\top)(\frac{\tau}{\tau+\sigma^2})^{j-1} + \sigma^2} \leq \frac{\lambda(\boldsymbol{X}\boldsymbol{X}^\top)}{\sigma^2}(\frac{\tau}{\tau + \sigma^2})^{j-1}.$$

$\square$

## A.2 DETAILED EXPLANATION OF THE GAP BETWEEN DIFFUSION MODELS AND DAES

In this appendix, we delve deeper into the connection between diffusion models and sequences of Denoising Autoencoders (DAEs), focusing on the initial step of the diffusion process.

Consider a diffusion model $f_\theta(t, \boldsymbol{x}_t)$ with $T$ time steps (e.g., $T = 1000$), which begins the sampling process from pure Gaussian noise $\boldsymbol{x}_0 \sim \mathcal{N}(0, \mathbf{I})$. The model predicts the target state using (here we consider the image $x$-prediction which is equal to noise $\epsilon$-prediction and velocity $v$-prediction (Salimans & Ho, 2022)):

$$\boldsymbol{x}_1 = f_\theta(0, \boldsymbol{x}_0), \tag{16}$$

where $f_\theta(0, \boldsymbol{x}_0)$ approximates the denoising function at time $t = 0$. This step functions as a DAE with pure Gaussian input.

Subsequent sampling steps involve Euler updates of the form:

$$\begin{aligned} \boldsymbol{x}_{0+\gamma} &= \boldsymbol{x}_0 + \gamma \left( f_\theta(0, \boldsymbol{x}_0) - \boldsymbol{x}_0 \right) \\ &\cdots \\ \boldsymbol{x}_{t+\gamma} &= \boldsymbol{x}_t + \gamma \left( f_\theta(t, \boldsymbol{x}_t) - \boldsymbol{x}_t \right), \end{aligned} \tag{17}$$

where $\gamma$ is a small time increment. In these steps, each input $\boldsymbol{x}_t$ is a mixture of Gaussian noise and previous model outputs, aligning with the typical input to a DAE trained on such mixtures.

The only significant gap between a sequence of DAEs and the diffusion model arises in the initial step due to the pure Gaussian input. By analyzing the initial step separately, we can better align the recursive DAE framework with the diffusion model. Specifically, if we consider the initial DAE handling pure Gaussian inputs and subsequent DAEs processing mixtures of noise and signal, the entire diffusion process can be viewed as a series of DAEs with varying input distributions.

However, an important question arises: *Will a linear DAE learn any meaningful information from the first step with pure Gaussian input?* In the case of a linear DAE, learning from pure noise is challenging because there is no underlying structure to capture. This limitation highlights why the initial step differs from the rest and underscores the necessity of separating its analysis.

By acknowledging this gap, our analysis becomes more comprehensive, bridging the understanding between DAEs and diffusion models. This perspective not only sheds light on the mechanics of diffusion models but also provides a pathway for leveraging insights from DAE analysis to improve diffusion-based generative models.

## A.3 PROOF OF PROPOSITION 2

Now, we formulate the reflow process of a linear DAE incorporating real data. Recall the settings from 4.2.1; suppose we have training data $\boldsymbol{X} = [\boldsymbol{x}_1 \quad \cdots \quad \boldsymbol{x}_n]$ with $\boldsymbol{x}_i = \boldsymbol{U}^\star \boldsymbol{U}^{\star\top} \boldsymbol{a}_i$, where $\boldsymbol{a}_i \sim \mathcal{N}(0, \mathbf{I})$. Starting with $\boldsymbol{X}_1 = \boldsymbol{X}$, the scheme for generating synthetic data at the $j$-th iteration ($j \geq 1$) is outlined as follows.

- **Add real data**: $\hat{\boldsymbol{X}}_j = [\boldsymbol{X}_j \quad \boldsymbol{X}]$.

- **Fit DAE**: $(\boldsymbol{W}_2^j, \boldsymbol{W}_1^j) = \theta^\star(\hat{\boldsymbol{X}}_j)$ by solving equation 8 with training data $\hat{\boldsymbol{X}}_j$.

- **Generate synthetic data for the next iteration**: $\boldsymbol{X}_{j+1} = \boldsymbol{W}_2^j \boldsymbol{W}_1^j (\boldsymbol{X}_j + \boldsymbol{E}_j)$, where each column of the noise matrix $\boldsymbol{E}_j$ is i.i.d. sampled from $\mathcal{N}(0, \hat{\sigma}^2/n^2 \mathbf{I})$.

First, we examine the effect of incorporating real data into the training process. Let $\lambda(\cdot)$ denote the largest eigenvalue of a matrix and $\lambda_{\min}(\cdot)$ denote the smallest eigenvalue of a matrix.

**Lemma A.1.** *Let $X_j, X_0 \in \mathbb{R}^{n \times d}$ be given matrices, and define the block matrix.*

$$\hat{\boldsymbol{X}}_j = [X_j \quad X_0].$$

*Then the maximum eigenvalue of $\hat{\boldsymbol{X}}_j \hat{\boldsymbol{X}}_j^\top$ satisfies the following inequalities:*

$$\lambda_{\min}(X_j X_j^\top) + \lambda(X_0 X_0^\top) \leq \lambda(\hat{\boldsymbol{X}}_j \hat{\boldsymbol{X}}_j^\top) \leq \lambda(X_j X_j^\top) + \lambda(X_0 X_0^\top).$$

*Proof.* First, observe that

$$\hat{X}_j \hat{X}_j^\top = X_j X_j^\top + X_0 X_0^\top. \tag{18}$$

We aim to bound $\lambda(\hat{X}_j \hat{X}_j^\top)$ using the eigenvalues of $X_j X_j^\top$ and $X_0 X_0^\top$. Recall that both $X_j X_j^\top$ and $X_0 X_0^\top$ are symmetric positive semi-definite matrices.

**Upper Bound:**

Using Weyl's inequality for eigenvalues of Hermitian matrices, we have

$$\lambda(A + B) \leq \lambda(A) + \lambda(B), \tag{19}$$

where $A$ and $B$ are symmetric matrices.

Applying this to $A = X_j X_j^\top$ and $B = X_0 X_0^\top$, we obtain

$$\lambda(\hat{X}_j \hat{X}_j^\top) \leq \lambda(X_j X_j^\top) + \lambda(X_0 X_0^\top).$$

**Lower Bound:**

Similarly, Weyl's inequality provides a lower bound:

$$\lambda(A + B) \geq \lambda_{\min}(A) + \lambda(B). \tag{20}$$

Applying this to $A = X_j X_j^\top$ and $B = X_0 X_0^\top$, we have

$$\lambda(\hat{X}_j \hat{X}_j^\top) \geq \lambda_{\min}(X_j X_j^\top) + \lambda(X_0 X_0^\top).$$

Combining the upper and lower bounds from Equations equation 19 and equation 20, we establish the inequalities in Equation equation A.1, thus proving the lemma. $\square$

**Proposition A.1.** *In the above synthetic data generation process 4.2.1 with adding real data, suppose that the variance of the added noise is not too large, i.e., $\hat{\sigma} \leq C\sigma$ for some universal constant $C$. Then, with probability at least $1 - 2je^{-n}$, the learned DAE does not suffer from model collapse as*

$$\|\boldsymbol{W}_2^j \boldsymbol{W}_1^j\|^2 \geq \frac{\|\boldsymbol{X}\|^2}{2\|\boldsymbol{X}\|^2 + \sigma^2}. \tag{21}$$

*Proof.* Following an analysis similar to the proof of Theorem 1, we have

$$\boldsymbol{\Phi}_j = (\hat{\mathbf{X}}_j \hat{\mathbf{X}}_j^\top) \left(\hat{\mathbf{X}}_j \hat{\mathbf{X}}_j^\top + \sigma^2 \mathbf{I}\right)^{-1}, \tag{22}$$

where $\hat{\mathbf{X}}_j = [\mathbf{X}_j \quad \mathbf{X}] \in \mathbb{R}^{n \times 2d}$. Since both $\boldsymbol{\Phi}_j$ and $\hat{\mathbf{X}}_j \hat{\mathbf{X}}_j^\top$ are symmetric positive semi-definite matrices, their eigenvalues are real and non-negative. Therefore, the eigenvalues of $\boldsymbol{\Phi}_j$ satisfy

$$\lambda(\boldsymbol{\Phi}_j) = \frac{\lambda(\hat{\mathbf{X}}_j \hat{\mathbf{X}}_j^\top)}{\lambda(\hat{\mathbf{X}}_j \hat{\mathbf{X}}_j^\top) + \sigma^2}. \tag{23}$$

Applying the eigenvalue bounds from Lemma A.1, we obtain

$$\lambda_{\min}(\hat{\mathbf{X}}_j\hat{\mathbf{X}}_j^\top) \geq \lambda_{\min}(\mathbf{X}_j\mathbf{X}_j^\top) + \lambda_{\min}(\mathbf{X}\mathbf{X}^\top), \tag{24}$$

$$\lambda(\hat{\mathbf{X}}_j\hat{\mathbf{X}}_j^\top) \leq \lambda(\mathbf{X}_j\mathbf{X}_j^\top) + \lambda(\mathbf{X}\mathbf{X}^\top). \tag{25}$$

Substituting these bounds into the expression for $\lambda_{\min}(\mathbf{\Phi}_j)$, we have

$$\lambda(\mathbf{\Phi}_j) \geq \frac{\lambda_{\min}(\mathbf{X}_j\mathbf{X}_j^\top) + \lambda(\mathbf{X}\mathbf{X}^\top)}{\lambda(\mathbf{X}_j\mathbf{X}_j^\top) + \lambda(\mathbf{X}\mathbf{X}^\top) + \sigma^2}. \tag{26}$$

Since $\lambda_{\min}(\mathbf{X}_j\mathbf{X}_j^\top) \geq 0$, it follows that

$$\lambda(\mathbf{\Phi}_j) \geq \frac{\lambda(\mathbf{X}\mathbf{X}^\top)}{\lambda(\mathbf{X}_j\mathbf{X}_j^\top) + \lambda(\mathbf{X}\mathbf{X}^\top) + \sigma^2}. \tag{27}$$

Let us denote $\tau = \lambda(\mathbf{X}\mathbf{X}^\top)$ and assume that $\lambda(\mathbf{X}_j\mathbf{X}_j^\top) \leq \tau$ (we will justify this assumption later). Then, we have

$$\lambda(\mathbf{\Phi}_j) \geq \frac{\lambda(\mathbf{X}\mathbf{X}^\top)}{2\tau + \sigma^2}.$$

Using a similar analysis as in the proof of Theorem 1, and the fact that $\mathbf{X}_{j+1} = \mathbf{\Phi}_j(\mathbf{X}_j + \mathbf{E}_j)$, where each column of $\mathbf{E}_j$ is independently sampled from $\mathcal{N}\left(0, \frac{\hat{\sigma}^2}{n^2}\mathbf{I}\right)$, we have

$$\lambda(\mathbf{X}_{j+1}\mathbf{X}_{j+1}^\top) \leq \lambda^2(\mathbf{\Phi}_j)\left(\lambda(\mathbf{X}_j\mathbf{X}_j^\top) + C\hat{\sigma}^2\right), \tag{28}$$

with probability at least $1 - 2e^{-n}$.

We will now prove that, with probability at least $1 - 2qe^{-n}$, the following holds:

$$\lambda(\mathbf{X}_q\mathbf{X}_q^\top) \leq \tau. \tag{29}$$

We proceed by induction. For $q = 0$, the inequality holds by the definition of $\tau$. Assume that inequality equation 29 holds for $q = j$; we will show it also holds for $q = j + 1$.

Since equation 29 holds at iteration $j$, we have $\lambda(\mathbf{X}_j\mathbf{X}_j^\top) \leq \tau$. Therefore,

$$\lambda(\mathbf{\Phi}_j) \leq \frac{\lambda(\mathbf{X}_j\mathbf{X}_j^\top) + \lambda(\mathbf{X}\mathbf{X}^\top)}{\lambda(\mathbf{X}_j\mathbf{X}_j^\top) + \lambda(\mathbf{X}\mathbf{X}^\top) + \sigma^2} \leq \frac{2\tau}{2\tau + \sigma^2}.$$

Plugging this bound, along with the assumption $\hat{\sigma}^2 \leq \frac{\sigma^2}{2C}$, into inequality equation 28, we obtain

$$\lambda(\mathbf{X}_{j+1}\mathbf{X}_{j+1}^\top) \leq \left(\frac{2\tau}{2\tau + \sigma^2}\right)^2 \left(\tau + C\hat{\sigma}^2\right) \leq \tau.$$

This completes the induction step and proves inequality equation 29.

Recall the inequality equation 27:

$$\lambda(\mathbf{\Phi}_j) \geq \frac{\lambda(\mathbf{X}\mathbf{X}^\top)}{\lambda(\mathbf{X}_j\mathbf{X}_j^\top) + \lambda(\mathbf{X}\mathbf{X}^\top) + \sigma^2}.$$

Since $\lambda(\mathbf{X}_j\mathbf{X}_j^\top)$ is bounded above by $\tau$ and $\lambda(\mathbf{X}\mathbf{X}^\top) > 0$, the right-hand side of inequality equation 27 is bounded below by a positive constant. Therefore, $\lambda(\mathbf{\Phi}_j)$ is bounded below by a positive constant, which implies that the learned DAE does not suffer from model collapse. $\qquad\square$

**Remark A.1.** *To prevent model collapse in generative models, a common strategy is to incorporate real data into the training process. Previous studies (Bertrand et al., 2023; Alemohammad et al., 2023; Gerstgrasser et al., 2024) have shown that mixing real data with synthetic data during training helps maintain model performance and prevents degeneration caused by relying solely on self-generated data. In diffusion models, integrating real samples can enhance model performance and reduce the risk of collapse (Kim et al., 2023). By conditioning the model on both real and synthetic data, the training process leverages the structure of real data distributions. Building on these approaches, our work introduces methods to integrate real data into the training of Rectified Flow, even when direct noise-image pairs are not available. By generating noise-image pairs from real data using reverse processes and balancing them with synthetic pairs, we prevent model collapse while retaining efficient sampling.*

### A.4 MODEL COLLAPSE IN RECTIFIED FLOW

In the appendix, we provide the formal statement of our proposition and the detailed proof:

*Proof.* Consider the explicit Euler discretization of the Rectified Flow ODE. Starting from $\boldsymbol{x}_{j,0} = \boldsymbol{z}$, where $\boldsymbol{z} \sim \mathcal{N}(0, \mathbf{I})$, we update:

$$\boldsymbol{x}_{j,t+\gamma} = \boldsymbol{x}_{j,t} + \gamma, v_{\theta_j}(t, \boldsymbol{x}_{j,t}), \quad t \in [0, 1], \tag{30}$$

with step size $\gamma$. If each small step of $v_{\theta_j}$ acts similarly to a DAE, then based on Theorem 1, as $j \to \infty$, we have:

$$\lim_{j \to \infty} \mathrm{rank}(v_{\theta_j}) = 0. \tag{31}$$

This implies $v_{\theta_j}(t, \boldsymbol{x}_{j,t}) \to \mathbf{0}$, leading to $\boldsymbol{x}_{j,t+\gamma} \approx \boldsymbol{x}_{j,t}$. Thus, the generated result remains near the initial point, confirming model collapse as stated in Proposition 1. $\square$

**Remark A.2.** *Although there is a theoretical gap between DAEs and Rectified Flow, our experimental results (Figure 6) support this proposition, suggesting that model collapse does occur in Rectified Flow under iterative self-training.*

## B METHODS DETAILS

### B.1 ONLINE REVERSE COLLAPSE AVOID REFLOW

---

**Algorithm 2** Online Collapse-Avoiding Reflow Training

---

**Require:** Reflow iterations $\mathcal{J}$; real dataset $\{\boldsymbol{x}^{(i)}\}$; pre-trained vector field $v_{\theta_0}$; mix ratio $\lambda$;
        SDE/ODE solver $\mathrm{SDE/ODE}(t_0, t_1, \cdot)$; regeneration parameter $\alpha$
**Ensure:** Trained vector fields $\{v_{\theta_j}\}_{j=1}^{\mathcal{J}}$
1: **for** $j = 1$ to $\mathcal{J}$ **do**
2:    **repeat**                                             ▷ *Reflow training*
3:       **for** each mini-batch **do**
4:          Sample $\{\boldsymbol{z}^{(i)}\}$ from $\mathcal{N}(\mathbf{0}, \mathbf{I})$
5:          Compute $\hat{\boldsymbol{x}}^{(i)} = \mathrm{SDE/ODE}(0, 1, \boldsymbol{z}^{(i)})$     ▷ *Generate synthetic data*
6:          Sample $\{\boldsymbol{x}^{(i)}\}$ from real dataset
7:          Compute $\hat{\boldsymbol{z}}^{(i)} = \mathrm{SDE/ODE}(1, 0, \boldsymbol{x}^{(i)})$     ▷ *Generate reverse data*
8:          Randomly select $\lambda B$ synthetic pairs and $(1 - \lambda)B$ real reverse pairs
9:          $\mathcal{D}_j = \{(\boldsymbol{z}_j^{(i)}, \boldsymbol{x}_j^{(i)})\} = \{(\boldsymbol{z}^{(i)}, \hat{\boldsymbol{x}}^{(i)})\} \cup \{(\hat{\boldsymbol{z}}^{(i)}, \boldsymbol{x}^{(i)})\}$    ▷ *Mix pairs according to $\lambda$*
10:         Sample $t \sim \mathcal{U}(0, 1)$
11:         **for** each $(\boldsymbol{z}_j^{(i)}, \boldsymbol{x}_j^{(i)})$ in $\mathcal{D}_j$ **do**
12:            Compute $\boldsymbol{x}_t^{(i)} = t\,\boldsymbol{x}_j^{(i)} + (1 - t)\,\boldsymbol{z}_j^{(i)}$
13:            Compute loss:

$$\mathcal{L}_{\mathrm{RF}} = \frac{1}{B} \sum_{i=1}^{B} \left\| v_{\theta_j}(t, \boldsymbol{x}_t^{(i)}) - (\boldsymbol{x}_j^{(i)} - \boldsymbol{z}_j^{(i)}) \right\|^2$$

14:         Update $\theta_j$ using gradient descent
15:    **until** converged
16: **Output:** $\{v_{\theta_j}\}_{j=1}^{\mathcal{J}}$

---

## B.2 Does Adding Randomness Help? Reverse SDE Sampling

In the previous methods, we utilized the reverse ODE process to generate noise-image pairs for training. However, when using only the deterministic ODE, the randomness in the training process originates solely from the initial latent variables $z \sim \mathcal{N}(0, I)$. This limited source of randomness may impact the diversity of the generated samples and the robustness of the model (Zhang et al., 2023b).

To enhance diversity and potentially improve generation quality, we consider introducing controlled randomness into the reverse process by employing a reverse Stochastic Differential Equation (SDE). The reverse SDE allows us to inject noise at each time step during the sampling process, defined as:

$$dx = \left[ f(t, x) - g(t)^2 \nabla_x \log p_t(x) \right] dt + g(t) d\widetilde{w}, \tag{32}$$

where $f(t, x)$ and $g(t)$ are the drift and diffusion coefficients, respectively, and $d\widetilde{w}$ denotes the standard Wiener process in reverse time. By introducing the diffusion term $g(t) d\widetilde{w}$, we inject controlled stochasticity into the reverse sampling.

In practice, we set the noise scale $g(t)$ to be small (e.g., $\sigma = 0.001$) and perform sampling using methods like the Euler-Maruyama scheme with an appropriate number of steps (e.g., 100 steps). This controlled injection of noise increases the variability in the training data without significantly disrupting the straightening effect of the flow. Specifically, the added randomness helps explore the neighborhood of data samples, enriching the learning process. We denote this method as **OCAR-S**.

## C  Experiments Details and Extra Results

### C.1  Gaussian Task

**Setup for DAE.** In the Reflow verification experiment for the DAE, we use a 4-dimensional Gaussian distribution as both the initial and target distributions. The initial distribution is $\mathcal{N}(0, I)$, and the target distribution is $\mathcal{N}(0, 5I)$, where $0$ is a 4-dimensional zero vector, and $I$ is the identity matrix. We employ a neural network $\theta$ composed of two linear layers $W_1$ and $W_2$ without activation functions and biases. We train the Reflow process for 20 iterations. The "Ratio" refers to the proportion of synthetic data to real data; a higher value indicates a greater proportion of synthetic data.

Figure 4 presents the results from the Reflow experiment using a Denoising Autoencoder (DAE) on a 4-dimensional Gaussian distribution.

**(a)** illustrates the rank of the weight matrix $W$ across different Reflow iterations. We set a threshold of $2 \times 10^{-1}$. Specifically, we perform Singular Value Decomposition (SVD) on $W_j$ and count the number of singular values greater than or equal to 0.2 to determine the rank of $W$. The results demonstrate that incorporating real data effectively prevents model collapse, as indicated by the maintenance of higher ranks. In contrast, relying solely on self-generated synthetic data leads to a rapid decline in rank towards zero.

**(b)** shows the Wasserstein-2 (W2) distance between the true target data distribution and the generated data distribution over Reflow iterations. This metric assesses the fidelity of the generated data in approximating the target distribution.

**(c)** displays the evolution of the first principal component (Dimension 0) of the data as Reflow iterations increase. We compare the original DAE, which does not utilize synthetic data, with our DAE-CA model, which employs various ratios of synthetic data (ranging from 0.1 to 0.9), as well as a fully synthetic DAE. The comparison highlights the effectiveness of our DAE-CA model in maintaining the integrity of principal components, thereby preserving data structure and diversity.

**Setup for Rectified Flow.** In the Reflow verification experiment for linear neural network Rectified Flow, we augment $W_1$ by adding one dimension corresponding to time, resulting in a neural network $W_1 W_2 : \mathbb{R}^{d+1} \to \mathbb{R}^d$. Our experimental results can be found in the below and confirm our prop 1 We also test a nonlinear neural network consisting of three linear layers with SELU activation functions and an extra dimension added to the first linear layer. The results are shown in

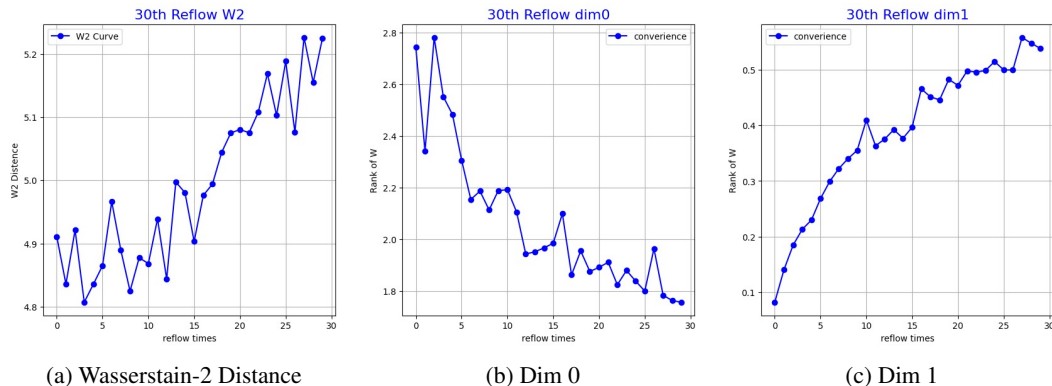

(a) Wasserstain-2 Distance    (b) Dim 0    (c) Dim 1

Figure 6: Results from the reflow experiment with linear Rectified flow on 10D Gaussian.

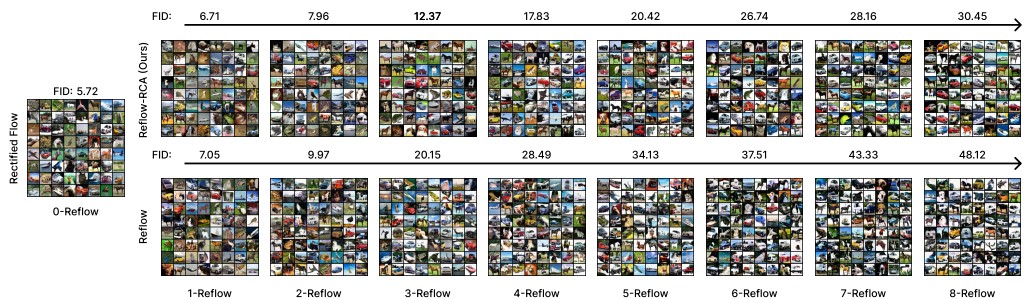

Figure 7: Results from the reflow experiment in CIFAR-10 using half-scale U-net.

## C.2    MODEL COLLAPSE IN LINEAR RCTIFIED FLOW

We experiment on a 10-dimensional Gaussian which starts from the initial distribution $\mathcal{N}(\mathbf{0}, \mathbf{I})$, and the target distribution is $\mathcal{N}(\mathbf{0}, 5\mathbf{I})$. But to demonstrate our inference, we set dimension 1 of the covariance matrix to 1e-3, which reduces the rank of the data as a whole. Figure 6a shows the model collapse process of linear RF, the Figure 6b and Figure 6c demonstrates the correctness of Propositio 1

## C.3    STRAIGHT FLOW AND FEWER-STEP IMAGE GENERATION

In our RCA Reflow experiments, due to the high computational cost of Reflow training, we use a half-size U-Net compared to the one used in Flow Matching (Lipman et al., 2022). For the qualitative experiments on CIFAR-10 shown in Table 2, we use a full-size U-Net with settings consistent with Lipman et al. (2022) to achieve the best performance. We used the standard implementation from the `https://github.com/atong01/conditional-flow-matching` repository provided by Tong et al. (2023). All methods were trained using the same configuration, differing only in the choice of the probability path or Reflow methods. Since the code for Lipman et al. (2022) has not been released, some parameters may still differ from the original implementation. We summarize our setup here; the exact parameter choices can be found in our source code. We used the Adam optimizer with $\beta_1 = 0.9$, $\beta_2 = 0.999$, $\epsilon = 10^{-8}$, and no weight decay. To replicate the architecture in Lipman et al. (2022), we employed the U-Net model from Dhariwal & Nichol (2021) with the following settings: channels set to 256, depth of 2, channel multipliers of [1, 2, 2, 2], number of heads as 4, head channels as 64, attention resolution of 16, and dropout of 0.0. We also used the "ICFM" methods from Tong et al. (2023)'s repository to train Rectified Flow instead of using the original repository open-sourced by Liu et al. (2022), because they use the same interpolation methods and probability paths.

Training was conducted with a batch size of 256 per GPU, using six NVIDIA RTX 4090 GPUs, over a total of 2000 epochs. For Reflow, we generated 500,000 noise-image pairs for every Reflow iteration, according to Liu et al. (2022)'s blog[1]. Although Liu et al. (2022) mention that they use 40,00,000 noise-image to get the best performance, we keep the regular 500,000 noise-image to save time and training source. The learning rate schedule involved increasing the learning rate linearly from 0 to $5 \times 10^{-4}$ over the first 45,000 iterations, then decreasing it linearly back to 0 over the remaining epochs. We set the noise scale $\sigma = 10^{-6}$. For sampling, we used Euler integration with the `torchdyn` package and the DOPRI5 solver from the `torchdiffeq` package.

Table 3: Summary of Configuration Parameters Across Experiments

|  | CIFAR10-figure 1 | CIFAR10-figure 5 | CIFAR10-Table 2 |
|---|---|---|---|
| Channels | 256 | 128 | 256 |
| Channels multiple | 1,2,2,2 | 1,2,2,2 | 1,2,2,2 |
| Heads | 4 | 4 | 4 |
| Heads Channels | 64 | 64 | 64 |
| Attention resolution | 16 | 16 | 16 |
| Dropout | 0.0 | 0.0 | 0.0 |
| Effective Batch size | 256 | 256 | 256 |
| GPUs | 6 | 6 | 6 |
| Noise-image pairs | 100k | 500k | 500k |
| Reflow Sampler | dopri5 | Euler (100 NFE) | dopri5 |
| $\alpha$ | 2 | 4 | 2 |
| $\lambda$ | 0.1 | / | 0.5 |
| Learning Rate | 2e-4 | 5e-4 | 5e-4 |

In the CelebA-HQ experiments, we maintain the image resolution at $256 \times 256$. We utilize a pretrained Variational Autoencoder (VAE) from Stable Diffusion (Rombach et al., 2022), where the VAE encoder reduces an RGB image $\mathbf{x} \in \mathbb{R}^{h \times w \times 3}$ to a latent representation $\mathbf{z} = \mathcal{E}(\mathbf{x})$ with dimensions $\frac{h}{8} \times \frac{w}{8} \times 4$. We used the standard implementation from the LFM repository (`https://github.com/VinAIResearch/LFM`) provided by Dao et al. (2023). We also used the DiT-L/2 (Peebles & Xie, 2023) checkpoint released in Dao et al. (2023)'s repository as the starting point for our Reflow training. Training was conducted with 4 NVIDIA A800 GPUs.

For RCA Reflow, we tested $\lambda \in 0.1, 0.3, 0.5, 0.7, 0.9, 1.0$ with $\alpha = 4$. Note that when $\lambda = 0.0$, we are using 100% real reverse image-noise pairs, which is not equivalent to the original Reflow of Rectified Flow. Therefore, we train the original Reflow as the baseline. For the regeneration parameter $\alpha$, we fixed $\lambda = 0.5$ and compared $\alpha \in 2, 4, 10, \infty$, where $\infty$ means we never regenerate new data within a single Reflow training. We evaluated the models using both the adaptive sampler "dopri5" (consistent with Lipman et al. (2022)) and fixed, low numbers of function evaluations (NFEs) $10, 20, 50$ to demonstrate the elimination of model collapse and the maintenance of flow straightness by our method. This allows us to assess both generation quality and sampling efficiency simultaneously.

## C.4 EXTRA RESULTS

**Parameter Ablation** Here we set the same setting in table 3 column 2.

Table 4: Performance of RF-RCA Models under Different $\lambda$ Values

| $\lambda$ | 0.1 | 0.3 | 0.5 | 0.7 | 0.9 | 1.0 |
|---|---|---|---|---|---|---|
| 1-RF-RCA | 5.87 | 6.21 | 6.37 | 6.81 | 6.93 | 7.05 |
| 2-RF-RCA | 6.37 | 7.10 | 7.96 | 8.53 | 8.98 | 9.97 |
| 3-RF-RCA | 8.02 | 10.29 | 12.37 | 14.74 | 18.01 | 20.15 |

---

[1] `https://zhuanlan.zhihu.com/p/603740431`

Table 5: Performance of RF-RCA Models under Different $\alpha$ Values

| $\alpha$ | 2 | 4 | 8 | $\infty$ |
|---|---|---|---|---|
| 1-RF-RCA | 6.09 | 6.37 | 6.70 | 7.05 |
| 2-RF-RCA | 6.92 | 7.10 | 8.14 | 9.97 |
| 3-RF-RCA | 9.71 | 10.29 | 13.37 | 20.15 |

**Precision and Recall** Here we set the same setting in table 3 column 2.

Table 6: Precision and Recall Performance on CIFAR10 and CelebA-HQ Datasets

| Precision/Recall | CIFAR10 | CelebA-HQ |
|---|---|---|
| 0-RF | 0.652 / 0.594 | 0.863 / 0.610 |
| 1-RF | 0.667 / 0.556 | 0.857 / 0.514 |
| 1-RF-RCA | 0.658 / 0.587 | 0.859 / 0.549 |
| 2-RF | 0.673 / 0.528 | 0.872 / 0.436 |
| 2-RF-RCA | 0.661 / 0.563 | 0.867 / 0.501 |

**1/2 step results for CIFAR10**

Table 7: Performance of RF-RCA models under different NFEs. Original data from the cited papers are provided in brackets when available. We set $\lambda = 0.5$, $\alpha = 2$, and use full-scale U-Net for CIFAR-10.

| $NFE$ | 1 | 2 |
|---|---|---|
| 0-RF | 351.79 (378) | 154.65 |
| 1-RF | 15.27 (12.21) | 11.49 |
| 2-RF | 19.27 (8.15) | 17.57 |
| 1-RF-RCA | 12.27 | 10.89 |
| 2-RF-RCA | 16.04 | 14.99 |

