# OpenReview forum: "Model Collapse Analysis and Improvement for Rectified Flow Models"
_ICLR.cc/2025/Conference — Submitted to ICLR 2025_

### Official Review · Reviewer_uv1X · 2024-10-23

**Soundness:** 1
**Presentation:** 1
**Contribution:** 2
**Rating:** 3
**Confidence:** 5

**Summary:**

The paper analyses model collapse in rectified flow models, i.e. models that train on data-noise couplings induced by themselves. The main observation is that with the number of such reflow iterations the quality of the generated data decreases. The authors provide a theoretical investigation of the phenomena for the simplistic case of linear denoising autoencoders and based on that they propose several schemes of incorporating the original real data in the subsequent reflow iterations to prevent model collapse. The main idea is to map the original data to noise using the reverse ODE/SDE of the learned vector field. Experimentally, the authors justify the claims on gaussian data.

**Strengths:**

The motivation delivered in the introduction is clear. The investigated problem is interesting and the approach presented in the paper is, to the best of my knowledge, original. The authors also provide theoretical analysis of the problem.

**Weaknesses:**

The main weakness of the paper is the presentation. Frankly, the paper seems to be quite raw: there are a lot of typos, poor formatting of the equations (e.g. Equation 5), undefined variables (e.g. $\Phi$ in Line 264, $U^*$ in Line 278 or $E_j$ in Line 282), broken references to the results (e.g. Line 310, 340, 372, 417, 514), missing results (experiments on CelebA were claimed, but never presented in the paper). Besides this, most of the evaluation is limited to only toy data (Gaussian-to-Gaussian mapping) and the results on real data (CIFAR10) are mixed (see Questions). Because of this, and despite the listed strengths, I cannot vote for acceptance of the paper.

**Questions:**

Here are some questions to the authors and further concerns that influenced my decision:
- The connection between the theoretical analysis and the proposed method is unclear. How does integrating real data in the training help breaking the bound in Theorem 1? Appendix A.3 is the closest to discussing this, but given the poor presentation quality, it seems to be isolated from the rest of the paper.
- The mixing scheme in Equation 11 is questionable. As far as I understand, the pairs are created by mixing independent synthetic and real data with a convex interpolation. In principle, this can destroy the target distribution. Could the authors provide more details regarding this? Another interpretation could be training on both pairs without mixing them, but maybe with balancing the ratio between synthetic and real data, if needed.
- The FID on CIFAR from Figure 1 (4.67 at 10th Reflow-RCA) contradicts with the FID in Figure 5 that seems to be above 35. Could the authors clarify this?
- How do the performances of the full model on real data differ with and without RCA? Based on the results presented in Figure 5, RCA is also prone to model collapse, although not as much as the vanilla Reflow. So there is a question, whether the model actually benefits from more than 1-3 Reflow iterations. If not and if the decrease in quality due to model collapse is not significant for the first couple of Reflow iterations, then the advantages of using RCA are unclear.

---

> ### Author Response · Authors · 2024-11-21
> **Reply to Reviewer uv1X**
>
> Thank you very much for acknowledging the theoretical and methodological contributions of our paper. We apologize for the oversight in the rushed writing and submission of the experimental section. In the revised version, we have supplemented all the missing content, making the experimental section—especially the image experiments—clearer and more complete. We have also adjusted some of the appendices and the order of the main text to improve readability.
>
> We would like to address your specific questions and concerns:
>
> 1. **The connection between the theoretical analysis and the proposed method is unclear.**
>
>     We have reformulated the relevant proof in **Appendix A.3** and summarized it as **Proposition 2** in the main text. Under mild conditions, we demonstrate that incorporating real data provides a theoretical lower bound on the maximum eigenvalue of the Denoising Autoencoder (DAE) trained in each iteration. This theoretical insight explains how integrating real data into the training process helps to break the bound established in **Theorem 1**, thus improving the stability and performance of the model.
>
> 2. **The mixing scheme in Equation 11 is questionable.**
>
>     We appreciate your careful observation and apologize for the confusion caused by a typographical error in the original manuscript. In the revised version, see **Equation 12**, we clarify that we actually train using a proportional mixture of noise-image pairs generated in the forward process and image-noise pairs generated by the reverse flow. Specifically, we randomly sample and combine these pairs during training, as detailed in **Algorithm 1**. This approach avoids disrupting the target distribution, as we do not perform convex interpolation between independent synthetic and real data. Instead, we maintain the integrity of the data distributions while balancing the contributions of synthetic and real data.
>
> 3. **The FID on CIFAR from Figure 1 contradicts with the FID in Figure 5.**
>
>     We apologize for the inconsistency. As noted in **Appendix C.3**, **Table 3**, we provide detailed parameter descriptions for our experiments. It is important to note that the figure on the first page was intended as a schematic illustration, and we did not perform standard parameter Reflow training for that figure. To prevent misunderstanding, we have removed the FID values from that figure in the revised manuscript.
>
> 4. **Performance differences with and without RCA.**
>
>     Based on **Figure 5** and **Table 2**, we can clearly see that the Reflow-RCA method improves the performance of Reflow by mitigating model collapse. While we acknowledge that completely eliminating model collapse is very challenging, our method significantly enhances the generation quality during Reflow training. This leads to a better trade-off between generation efficiency and quality, effectively pushing the Pareto frontier forward. Notably, our **OCRA** and **OCRA-S** methods outperform all existing Reflow methods on real datasets, demonstrating the effectiveness of our approach.
>
>     Regarding your concern about the benefits of more than 1–3 Reflow iterations, our experiments show that while vanilla Reflow suffers from severe model collapse with increased iterations, our RCA method substantially reduces this effect. This allows for more iterations without significant degradation in performance, meaning that the model continues to benefit from additional Reflow iterations when using RCA, leading to improved results over the baseline.
>
> Furthermore, our method is compatible with many techniques aimed at improving the quality of generative models. For example, we can incorporate the LPIPS loss used in [1], the Pseudo-Huber loss from [2], or the specialized probability flow design in [3]. Additionally, our approach can be applied to existing simulation-free generative models with higher generation quality, such as [3], to achieve even greater generation efficiency.
>
> **References:**
>
> [1] Consistency Models, ICML'23.
>
> [2] Improved Techniques for Training Consistency Models, ICLR'24
>
> [3] Scaling rectified flow transformers for high-resolution image synthesis, ICML'24

---

> > ### Comment · Reviewer_uv1X · 2024-11-22
> > **Response to the Authors**
> >
> > Dear Authors,
> >
> > Thank you for your time and valuable additional clarifications in the rebuttal. The presentation has improved in the revised version of the paper. Because of this, I raised my rating.
> >
> > However, in my view, the current version substantially differs from the original submission and can be considered as a major revision. Besides this, I still have concerns and the presentation can be further improved. E.g. from Figure 5 it seems that there is no need in performing Reflow more than once. Reflow is used to straighten the sampling paths and decrease NFE, which makes 100-step evaluations irrelevant. And from the figure alone the significance of the improvements is not obvious for Reflow-1 and 1 or 2 steps. Moreover, I believe the results in Table 2 need more explanations. E.g. where does the drop in the performance compared to Liu et. al come from (Liu et. al report 1-step FID of 12.21 for Reflow-2, while in Table 2 the 1-step FID is 14.97)?
> >
> > Hence, I am still leaning towards rejecting the paper in the current form.
> >
> > Best regards,
> > Reviewer

---

> > > ### Author Response · Authors · 2024-11-22
> > > **Reply to Reviewer uv1X (References)**
> > >
> > > References:
> > >
> > > [1] Improving and generalizing flow-based generative models
> > > with minibatch optimal transport, TMLR'24
> > >
> > > [2] Flow matching for generative modeling, ICLR'23
> > >
> > > [3] Score-Based Generative Modeling through Stochastic Differential Equations, ICLR'21
> > >
> > > [4] Diffusion models beat gans on image synthesis, Neurips'2021
> > >
> > > [5] Improving the Training of Rectified Flows, Neurips'24
> > >
> > > [6] InstaFlow: One Step is Enough for High-Quality Diffusion-Based Text-to-Image Generation, ICLR'24
> > >
> > > [7] Rectified Flow: A Marginal Preserving Approach to Optimal Transport, Arxiv

---

> > > ### Author Response · Authors · 2024-12-02
> > > **Reply to Reviewer uv1X**
> > >
> > > Dear Reviewer uv1X，
> > >
> > > We would like to sincerely thank you for your constructive feedback. Based on your insightful comments, we have made several revisions to the manuscript that we believe enhance its clarity and rigor. We would like to emphasize that these revisions do not alter the core contributions, methodologies, or conclusions of the paper. Rather, the revisions provide additional clarification and detailed explanations to improve the presentation and transparency of the work.
> > >
> > > These revisions aim to address your concerns and provide a clearer understanding of the performance improvements, without changing the logical framework or the fundamental conclusions of the paper. We believe the updated manuscript presents our work more transparently and effectively.
> > >
> > > Thank you once again for your time and thoughtful feedback. We greatly appreciate your consideration of the revised manuscript.
> > >
> > > Best regards,
> > >
> > > Authors of Paper 5606

---

> ### Author Response · Authors · 2024-11-22
> **Reply to Reviewer uv1X**
>
> Dear Reviewer,
>
> Thank you for your response. Due to space limitations, we did not provide detailed descriptions of our parameter settings in the main text, causing you inconvenience, and we apologize for that.
>
> **Firstly**, the main purpose of our experiments is to verify the correctness of our theory in both Gaussian and image experiments and to demonstrate that RCA and OCRA can effectively alleviate the model collapse phenomenon during Reflow in Rectified Flow. We did not claim any state-of-the-art (SOTA) results. It is especially worth noting that **our method is compatible with most training techniques that can improve model performance**, such as the LPIPS loss mentioned in our previous response.
>
> **Secondly**, in all our self-trained comparative experiments, we ensured fair parameter settings. Unless new parameters were introduced, we maintained the parameters stated in the Flow Matching paper [1], [2] and ensured parameter consistency across the same experiments. Based on this, we believe our experiments are fair. We will provide specific explanations below.
>
> As shown in Figure 5(a), our RCA provides improvements in generation quality regardless of the number of sampling steps. Since we needed to perform 8 Reflow iterations to plot Figure 5—to showcase the model collapse phenomenon in Reflow and the improvements of our method—we had to adjust the parameters (e.g., using a half-scale U-Net, larger λ, and larger α) to **reduce training resource consumption** due to limited computational resources. **We believe this is the reason why the effect in Figure 5(a) is not as pronounced at 1–2 steps**, which is also reflected in our parameter ablation experiments (see Appendix C.4). Meanwhile, we provide very detailed parameter settings and the reasons for using these parameters in Appendix C.3. For better parameter tuning to demonstrate the advantages of RCA and OCRA, please refer to the experiments in Table 2, which better showcase our quantifiable advantages.
>
> In the quantitative experiments in Table 2, we used a full-scale U-Net (with parameters consistent with [2]) and a more optimized α, as shown in Appendix Table 3. It is worth mentioning that we used **500k noise-image pairs** because, according to the authors' blog (see our revision at line 1137 and footnote on page 22: https://zhuanlan.zhihu.com/p/603740431 ), in their response on 2023/02/07, they mentioned:
>
> "On CIFAR-10, we probably need to generate at least 500,000 pairs of (noise, image) to train 2-Rectified Flow with FID < 5; to achieve the results in the article, we generated 4 million pairs of (noise, image)." (Translated by Google)
>
> Since the comparative experiments for Reflow and Reflow-RCA are fair, we used 500k pairs to **save computational resources**. The training of 10-Reflow will take more than 300 hours using 6 GPUs.
>
> Furthermore, in all our experiments on CIFAR-10, we used the **codebase https://github.com/atong01/conditional-flow-matching**, which is provided by [1]. This is because Flow Matching [2] does not provide publicly available source code, and this codebase integrates most of the well-known PyTorch implementations of Flow Matching and Rectified Flow. According to our observations, the neural network used in the [1] codebase has some minor differences from the Rectified Flow provided source code. The Rectified Flow source code utilizes PyTorch/TensorFlow/JAX code styles and uses **NCSN** inherited from [3], while the network in [1] is based on [4]. Based on this, we retrained 0-Rectified Flow, and Table 2 shows our results. **To avoid misunderstanding, we also provide the values reported by the authors in the "Best NFE" column of the table**.
>
> Finally, we believe that although Rectified Flow **does not utilize many advanced training and generation techniques**—as partly reported by [5]—it may not match the most advanced distilled diffusion models, such as Consistency Models, in the 1–2 step setting. However, improvement techniques based on Rectified Flow remain meaningful because **straighter flows bring significant benefits, such as ease of distillation [6]**. We believe our method provides **new directions** for advancement.
>
> Moreover, we consider that multiple Reflow iterations have significant mathematical significance, as they offer a **theoretically rigorous approach to approximate optimal transport [7]**. This effectively demonstrates the combination of theory and practice. Our focus should be on optimizing the methods, such as exploring better ways to avoid model collapse in future work.
>
> We will include a more comprehensive **Table 2** in the appendix of the revision, which will contain 1–2 step sampling results to better reflect our advantages. Additionally, we will add a more detailed description of the experimental section in the revision to eliminate any misunderstandings.
>
> Once again, thank you for your careful reading. Your dedication plays a crucial role in driving the continuous development of our community.

---

### Official Review · Reviewer_dE32 · 2024-11-02

**Soundness:** 2
**Presentation:** 1
**Contribution:** 2
**Rating:** 6
**Confidence:** 3

**Summary:**

The paper addresses the issue of k-rectified flow, specifically focusing on the problem of drift that eventually leads to mode collapse. According to the authors, the root cause of this issue is that reflow relies solely on synthetic data, causing the generated distribution to drift too far from the original real image distribution over time. This insight stems from a theoretical framework based on Denoising Autoencoders. To mitigate this, the authors propose incorporating real images into the reflow process. By applying inversion on these images, they obtain inverted noise-real image pairs that can be integrated into the reflow process (RCA). Additionally, to improve efficiency, they introduce an online version that generates and mixes inverted noise-real image pairs dynamically.

**Strengths:**

The paper is technically rigorous, with a strong foundation in theoretical analysis that clearly leads to practical implementation. Using Denoising Autoencoders (DAEs) to illustrate mode collapse is particularly effective, as it provides a helpful visualization of the issue. The topic (mode collapse) is a fundamental limitation in k-rectified flow models, and addressing it could have significant impact on improving model stability. The proposed methods offers a well-rounded solution and quite efficient.

**Weaknesses:**

Presentation:
- The presentation lacks thoroughness and one major part is put into the supplementary material.
- A lot of missing/ undefined references
Experiments:
- Not in main paper
- A lot of promised results are not presented, all the outcome is only on toy samples (gaussian dataset). All the experiment upon real dataset such as upon CIFAR-10, CelebA-HQ are mentioned however no where to be found. Also lacks of metrics to check about mode collapse such as recall.
- Also no qualitative results upon the real datasets.
- Lack of ablation studies upon several aspects: amount of real dataset samples, mix ratio $\lambda$,...

**Questions:**

The author should address problems mentioned in weakness.

---

> ### Author Response · Authors · 2024-11-21
> **Reply to Reviewer dE32**
>
> Thank you sincerely for acknowledging the theoretical and methodological contributions of our paper. We apologize for the oversight in the experimental section caused by our rushed writing and submission process. In an effort to meet the 10-page limit, we mistakenly omitted some experimental tables and descriptions. In the revised version, we have restored all missing content, making the experimental section—especially the image experiments—more comprehensive and clearer. We have also adjusted the order of some appendices and the main text to improve coherence.
>
> We would like to clarify an important point regarding the **model collapse** phenomenon studied in our paper, which differs from **mode collapse**. Mode collapse primarily occurs in GANs and refers to the generative model producing only a few modes or samples from the data distribution while neglecting others. In contrast, model collapse refers to the progressive degradation in performance of generative models trained iteratively on their own outputs. In our revision, we have included images generated by models trained with k-reflow that illustrate this gradual deterioration, helping to better understand model collapse. We apologize for any confusion this may have caused.
>
> > **Weaknesses 1: Presentation**
>
> Regarding the areas that needed improvement:
>
> 1. **Restored Table 2**: Located at line 512 in the main text.
>
> 2. **Redrawn Figure 5**: We have updated the figure to present clearer visualization curves and included additional generated results to enhance understanding.
>
> 3. **Precision/Recall Measurements**: We greatly appreciate your suggestion to test recall. We conducted precision and recall evaluations, and the results are provided in **Appendix C.4**. Our findings indicate that iterative Reflow training reduces the recall of generated images, implying decreased diversity. Our RCA-Reflow method mitigates this decline effectively.
>
> 4. **Sample Images on CIFAR-10**: Provided in **Figure 5(b)** to illustrate our results more concretely.
>
> 5. **Ablation Study on RCA Parameters**: We included an ablation study of the RCA method's parameters, with results presented in **Appendix C.4**.

---

> > ### Comment · Reviewer_dE32 · 2024-11-25
> >
> > Thank you, authors, for carefully addressing my concerns and revising the paper. I am pleased to note that the latest version successfully resolves the issues I had previously raised, and as a result, I have raised my score.
> >
> > That said, I find myself in agreement with reviewer uv1X. Given the significant scope of the revisions, the current manuscript feels more akin to a new submission rather than a continuation of the original one.

---

> > > ### Author Response · Authors · 2024-11-25
> > > **Reply to Reviewer dE32**
> > >
> > > Dear Reviewer dE32,
> > >
> > > Thank you for your thoughtful feedback and for taking the time to reassess our manuscript. We appreciate your recognition that our revisions have addressed your initial concerns, leading to an improved evaluation.
> > >
> > > Regarding your observation that the revised manuscript resembles a new submission, we would like to clarify that our revisions were strictly in response to your specific feedback. Specifically:
> > >
> > > **Presentation:**
> > >
> > > >Lack of Thoroughness and Supplementary Material:
> > >
> > > Response: We have relocated the previously supplementary material into the main manuscript to enhance the thoroughness of our presentation. This ensures that all major components of our work are readily accessible within the primary document, providing a more cohesive and comprehensive narrative.
> > >
> > > >Missing/Undefined References:
> > >
> > > Response: We have conducted a thorough review of our manuscript to identify and rectify all missing or undefined references. Each reference has been carefully checked for accuracy and completeness. Additionally, we have ensured that all citations are appropriately linked to their corresponding bibliography entries, thereby improving the overall clarity and reliability of our references section.
> > >
> > > **Experiments:**
> > >
> > > >Inclusion of Promised Results and Real Datasets:
> > >
> > > Response: We have added the previously promised experimental results on real-world datasets, specifically CIFAR-10 and CelebA-HQ, to the main body of the paper. These results are now presented in Section 6.2. By incorporating these experiments into the main manuscript, we provide a more robust and comprehensive evaluation of our proposed approach beyond the initial toy Gaussian dataset.
> > >
> > > >Additional Metrics to Assess Mode Collapse (e.g., Recall):
> > >
> > > Response: To address concerns regarding model collapse, we have introduced additional evaluation metrics, including recall, into our experimental analysis. The inclusion of recall alongside existing metrics offers a more nuanced understanding of our model's effectiveness in mitigating mode collapse.
> > >
> > > >Qualitative Results on Real Datasets:
> > >
> > > Response: We have included qualitative results for the real-world datasets, CIFAR-10 and CelebA-HQ, in Section 4.5. These qualitative analyses provide visual evidence of our model's performance and its ability to generate diverse and high-quality samples, thereby complementing the quantitative results previously presented.
> > > Ablation Studies on Various Aspects:
> > >
> > > **Summary of Revisions:**
> > >
> > > 1. Relocated major sections from supplementary material to the main text for improved thoroughness.
> > >
> > > 2. Added more comprehensive experimental results on CIFAR-10 and CelebA-HQ within the main manuscript.
> > >
> > > 3. Introduced additional metrics, including recall, to assess model collapse.
> > >
> > > 4. Provided qualitative results for real datasets to complement quantitative analyses.
> > >
> > > 5. Conducted and presented ablation studies on key aspects influencing model performance.
> > >
> > > 6. Ensured all references were complete and correctly defined throughout the manuscript.
> > >
> > > These revisions were meticulously implemented to directly **address your valuable feedback**, enhancing both the presentation and the empirical evaluation of our work **without altering the core contributions or introducing new methodologies**. Our objective was to ensure that the manuscript is as clear, thorough, and robust as possible, in alignment with the ICLR submission guidelines.
> > >
> > > We hope this clarifies that our revisions are in line with the expectations for the rebuttal phase as outlined by ICLR. Thank you once again for your valuable insights and for contributing to the improvement of our manuscript.
> > >
> > > Best regards,
> > >
> > > Authors of Paper 5606

---

### Official Review · Reviewer_NPGT · 2024-11-04

**Soundness:** 3
**Presentation:** 2
**Contribution:** 3
**Rating:** 5
**Confidence:** 3

**Summary:**

This paper proposes to avoid model collapse in rectified flow model for generative modeling. From theoretical analysis, this paper shows that the training of rectified flow model is affected by the training samples and proposes reverse collapse-avoiding reflow by mixing synthetic and real reverse pairs. This paper shows a toy example on Gaussian data distribution, CIFAR-10, and CelebA-HQ256 dataset. Overall, this paper shows some point interesting to avoid model collapse in rectified flow model, however, it seems this paper is not ready for a strong submission. Therefore, I slightly lean to reject this paper, but may change my final rating after reading other reviewers' comments and authors' rebuttal.

**Strengths:**

+ This paper provides theoretical analysis for rectified flow model, and points out the connection/differences to diffusion models. The rectified model is more efficient in sampling than diffusion models.

+ According to the experiments with Gaussian distribution, this paper shows the effectiveness of the proposed solution.

**Weaknesses:**

+ Paper writing needs to improve. It seems this paper is completed in the rush and not ready for a strong submission. There are too many "?" in the Figure/Table/etc.
Line 310 Appendix?
Line 340 Figure ?
Line 372 Algorithm?
Line 417 Appendix ?
Line 514 Table ?

+ This paper claims the experiments are conducted for CIFAR-10 and CelebA-HQ256 datasets, however, the quantitative numbers and qualitative results are not shown in the main submission. Lacking of experiments is a fatal point and hard to make readers convinced.

+ As mentioned in the abstract and introduction, the proposed method can be used to generate high resolution images. However, 256x256 images are generated as reported, which is not enough. Higher resolution such as 1024x1024 will be more interesting.

**Questions:**

What is the relationship between \head{x}^{(i)} and x^{(i)}? Does the method sample x^{(i)} from the dataset randomly?

---

> ### Author Response · Authors · 2024-11-21
> **Reply to Reviewer NPGT**
>
> We sincerely appreciate your acknowledgment of the theoretical and methodological contributions of our paper. We apologize for the oversight in the experimental section, which was due to the rushed nature of our writing and submission process.
>
> > **Weaknesses 1: Paper Writing**
>
> We sincerely apologize for the rushed submission, which led to issues such as unresolved hyperlinks. These have all been corrected in the revised version. The specific updates you mentioned have been addressed as follows:
>
> 1. **Added content and clarifications:**
>     - **Line 310**: Appendix C1 and C2
>     - **Line 340**: Figure 2
>     - **Line 372**: Algorithm 1
>     - **Line 417**: Appendix B.2
>     - **Line 514**: Table 2
>
> > **Weaknesses 2: Extra Quantitative Numbers and Qualitative Results**
>
> 1. Regarding the **CIFAR-10 and CelebA-HQ256 datasets**, we regret that some content was mistakenly commented out during our adjustments to meet the page limit. We sincerely apologize for this oversight.
>
>     - In the revision, we have restored the table (**Table 2**) and have also re-plotted the model collapse curves for CIFAR-10 (**Figure 5** ) to provide a clearer visual comparison.
>     - Additionally, we included sample images illustrating the **model collapse process** and the effectiveness of our **RCA method** in mitigating this issue. These additions provide both quantitative and qualitative evidence to support our claims.
>
> > **Weaknesses 3: High-Resolution Images**
>
> We acknowledge the concerns regarding high-resolution image generation and address them as follows:
>
> 1. Our method is compatible with commonly used **diffusion-based high-resolution image generation techniques**, such as latent space modeling. This is evident in our experiments with **CelebA-HQ 256**.
>
> 2. While we agree that additional experiments with higher resolutions could enhance our results, we note that the complexity of our current experiments already surpasses most **model collapse** studies. For example:
>
>     - **[1]** and **[2]** use datasets with resolutions of only **32** or **64** pixels.
>     - Similarly, works in the generative modeling field, such as **[3]**, report experiments with datasets capped at **256** pixels.
> 3. Regarding **10-RF CIFAR-10 experiments** (Figure 5), we used a reduced channel size (**num of channels = 128**, which is half of the recommended size in **Flow Matching [5]**) to expedite training (details in Appendix C3). Despite these optimizations, training required over 7 days to complete **10 rounds of Reflow**.
>
> 4. Due to our limited computational resources, we were unable to extend our experiments to resolutions like **1024×1024**. However, our results with **latent space techniques** indicate that our approach is well-suited to **enhancing high-resolution image generation** performance in **Rectified Flow [4]**.
>
> #### **Question 1**
>
> As detailed in **Section 5.1** and **Algorithm 1**:
>
> 1. $x^{(i)}$ refers to a randomly sampled image from the training dataset, used to generate a matching noise vector $\hat{z}^{(i)}$.
> 2. $\hat{x}^{(i)}$ represents the generated image obtained by sampling with the random noise vector $z^{(i)}$.
>
> **References:**
>
> [1] On the Stability of Iterative Retraining of Generative Models on their own Data, ICLR'24.
>
> [2] Is Model Collapse Inevitable? Breaking the Curse of Recursion by Accumulating Real and Synthetic Data, COLT'24.
>
> [3] Consistency Models, ICML'23.
>
> [4] Scaling Rectified Flow Transformers for High-Resolution Image Synthesis, ICML'24.
>
> [5] Flow Matching for Generative Modeling, ICLR'23.

---

> > ### Author Response · Authors · 2024-12-02
> > **Reply to Reviewer NPGT**
> >
> > Dear Reviewer,
> >
> > Thank you for your thoughtful feedback on our paper. We have carefully addressed the concerns you raised in our rebuttal, and we hope the clarifications provided have resolved the issues to your satisfaction.
> >
> > As the rebuttal period is coming to a close, we would like to kindly ask if you would consider revising your score in light of the updates and explanations we have provided. We genuinely believe the revisions have strengthened the paper and would greatly appreciate any adjustments you could make to reflect this.
> >
> > We truly value your feedback and appreciate the time you’ve dedicated to reviewing our submission.
> >
> > Best regards,
> >
> > Authors of Paper 5606

---

### Author Response · Authors · 2024-11-21
**To all Reviewers**

We sincerely appreciate the time and effort you have dedicated to reviewing our paper. We have carefully considered your valuable feedback and have made comprehensive revisions throughout the manuscript. Detailed responses to your comments will be provided in the respective replies. We remain open to any additional suggestions you may have. If our revisions address your concerns satisfactorily, we would be truly grateful if you could consider raising our score.

---

### Author Response · Authors · 2024-12-04
**To all Reviewers**

Dear Reviewer,

We sincerely appreciate the time and effort you have dedicated to reviewing our paper. We have carefully considered your valuable feedback and have made revisions to the manuscript accordingly. These improvements fully comply with your suggestions and aim to enhance the clarity and presentation of our work, without altering the main contributions, statements, or methods.

As today is the final day for author responses, we kindly ask if our revisions have satisfactorily addressed your concerns. If you feel that we have effectively resolved the issues you raised, we would be truly grateful if you could consider adjusting your score.

Thank you again for your thoughtful feedback. Please let us know if you have any additional suggestions or questions.

Best regards,

Authors of Paper 5606

---

### Meta-Review · Area_Chair_ChFr · 2024-12-21

**Metareview:**

Summary: The paper investigates model collapse in Rectified Flow, focusing on the causes of performance degradation when using iterative self-generated training data.
Strengths: The problem of studying model collapse is fundamental to enhancing and understanding model stability. Although I am not a domain expert, the authors provide a sound theoretical analysis of the issue. The use of denoising autoencoders as an illustrative tool is effective, making the methodology accessible and well-scoped.
Weaknesses: While the authors have made significant efforts to address the reviewers' comments in their revisions, the technical explanations remain somewhat convoluted and could benefit from further simplification. At a high level, the objectives and proposals are clear, but the detailed presentation is challenging to follow. Additionally, concerns persist regarding experiments involving multiple Reflow operations and the presentation of experimental results, as highlighted by Reviewer uv1x.
Rejection Reason: See weakness and additional comments.

**Additional Comments On Reviewer Discussion:**

The paper received mixed ratings: 1x reject, 1x marginally below acceptance, and 1x marginally above acceptance.
After contacting all reviewers offline via email, the ratings have not changed. I am not an expert on this domain and I am at this point leaning towards rejecting, as the rebuttal has not also convinced me to overwrite the reviewers main concerns

---

### Decision · Program_Chairs · 2025-01-22

Reject